# Templated dewetting of single-crystal sub-millimeter-long nanowires and on-chip silicon circuits

Monica Bollani [1]*, Marco Salvalaglio [2]*, Abdennacer Benali[3], Mohammed Bouabdellaoui [3,4], Meher Naffouti[3,5], Mario Lodari[1], Stefano Di Corato[1], Alexey Fedorov[1], Axel Voigt [2,6], Ibtissem Fraj[3,5], Luc Favre [3], Jean Benoit Claude[3], David Grosso[3], Giuseppe Nicotra[7], Antonio Mio[7], Antoine Ronda[3], Isabelle Berbezier[3] & Marco Abbarchi [3]*

Large-scale, defect-free, micro- and nano-circuits with controlled inter-connections represent the nexus between electronic and photonic components. However, their fabrication over large scales often requires demanding procedures that are hardly scalable. Here we synthesize arrays of parallel ultra-long (up to 0.75 mm), monocrystalline, silicon-based nano-wires and complex, connected circuits exploiting low-resolution etching and annealing of thin silicon films on insulator. Phase field simulations reveal that crystal faceting and stabilization of the wires against breaking is due to surface energy anisotropy. Wires splitting, inter-connections and direction are independently managed by engineering the dewetting fronts and exploiting the spontaneous formation of kinks. Finally, we fabricate field-effect transistors with state-of-the-art trans-conductance and electron mobility. Beyond the first experimental evidence of controlled dewetting of patches featuring a record aspect ratio of ∼1/60000 and self-assembled ∼mm long nano-wires, our method constitutes a distinct and promising approach for the deterministic implementation of atomically-smooth, mono-crystalline electronic and photonic circuits.

[1] Istituto di Fotonica e Nanotecnologie-Consiglio Nazionale delle Ricerche, Laboratory for Nanostructure Epitaxy and Spintronics on Silicon, LNESS, Via Anzani 42, 22100 Como, Italy. [2] Institute of Scientific Computing, Technische Universität Dresden, 01062 Dresden, Germany. [3] Aix Marseille Univ, Université de Toulon, CNRS, IM2NP Marseille, France. [4] Laboratory of Physics of Condensed Matter and Renewable Energy, Faculty of Sciences and Technology, Hassan II University of Casablanca, 146 Mohammedia, Casablanca, Morocco. [5] Laboratoire de Micro-Optoélectronique et Nanostructures, Faculté des Sciences de Monastir, Université de Monastir, 5019 Monastir, Tunisia. [6] Dresden Center for Computational Materials Science, Technische Universität Dresden, 01062 Dresden, Germany. [7] CNR-IMM, Zona Industriale Strada VIII, 5, 95121 Catania, Italy. *email: monica.bollani@ifn.cnr.it; marco.salvalaglio@tu-dresden.de; marco.abbarchi@im2np.fr

Semiconductor nanowires exhibit superior and configurable electronic and optical properties[1,2]. Their disruptive potential has been demonstrated in photonics[3] (e.g. for lasers[4] or quantum optics[5]), electronics[6], thermoelectricity[7], energy storage with batteries[8], gas[9] and mechanical[10] sensing, topological quantum states[11], and much more. For these reasons their growth has been tackled with a plethora of techniques aiming at the production of controlled, ultra-long structures matching the needs of high yield, scalability (e.g. integration of a large number of devices on the same monolithic nano-wire) and material quality (e.g. smooth interfaces).

As such, the range of available approaches to grow elongated crystalline structures steadily increases[1], from direct top-down design, as, e.g., three-dimensional mesoscale lithography[12], super-lattice nanowire pattern transfer[13] up to the exploitation of natural phenomena such as the renowned Plateau-Rayleigh instability[14]. Several bottom-up, self-assembly methods can be employed to obtain high-quality parallel wires[15]. Nevertheless, a full control over their morphology, size, position, direction, inter-connection, and electrical isolation remains a challenge as current techniques are not versatile and are limited to a few micrometers length. On the other side, lithographic top-down methods can be used to precisely prepare a substrate for further engineering, as for instance shown for the cases of controlled wetting[16] or nano-imprint lithography[17]. Hybrid top-down/bottom-up approaches[18–21] can marry the crystalline quality of epitaxial self-assembly with the ultimate control of nano-structures position and size and create quantum-grade materials, such as complex assemblies of connected wires and membranes. However, their exploitation in scalable devices is often hindered by the complexity of their implementation requiring too many, cumbersome nano-fabrication steps. In fact, these approaches often need high-resolution etching (e.g. in order to set the final size of the epitaxial structures) and provide out-of-plane objects eventually requiring further processing before their implementation in a device[18,22]. These methods are hardly scalable as they exploit complex epitaxial growth steps, often involving a metallic catalyst and are limited to structures extending only over a few micrometers.

Here we fabricate arrays of in-plane, ultra-long nano-wires (up to 0.75 mm) and complex inter-connected circuits of mono-crystalline silicon using a catalyst- and epitaxy-free, hybrid top-down/bottom-up approach based on the natural morphological evolution of thin solid films. We control the metamorphosis of a commercial (001)-oriented ultra-thin silicon film on insulator (UT-SOI) relying only on low-resolution etching and annealing, directly transforming it in monocrystalline nano-wires. The final structures have a lateral size up to five times smaller than the etched patch width, are obtained with size- and position-control and are electrically-isolated from the substrate. Phase field simulations of surface-diffusion-limited kinetics elucidate the key role of the surface-energy anisotropy in stabilizing the dewetting outcome against breaking. They quantitatively reproduce the main features of the morphological evolution of the patches.

Exploiting the orientation-dependent edge faceting promoted by surface energy minimization[23] which hinders the onset of the typical Rayleigh-like instability along the wire axis[24] we extend these results to arbitrary in-plane crystallographic directions, building complex circuits of wires featuring splitting, changes in their directions and inter-connections. Finally, with a simple spin-on-dopant post-fabrication method, we render the wires conductive, demonstrating the possibility to use them as field-effect transistor[6] with trans-conductance and electron mobility similar to state-of-the-art nanowires devices.

## Results

### Templated dewetting along stable dewetting fronts.
Dewetting of monocrystalline thin silicon films is a spontaneous phenomenon where capillary forces drive mass transport via surface-diffusion-limited kinetics[25,26]. It leads to a complete metamorphosis of the flat layer in three-dimensional structures through hole nucleation, rims formation (where mass accumulates while receding), followed by finger-like structures and finally, in isolated islands. This natural shape evolution can, however, be controlled by engineering the dewetting fronts by patterning the UT-SOI prior to annealing[27,28].

Although silicon dewetting is also possible starting from an amorphous UT-SOI[29,30], the need of a precise and controlled dewetting front in order to form regular nano-architectures requires mono-crystalline wafers where the dewetting process only affects the shape of the layer which always remains a mono-crystal[31–33]. The potential of templated dewetting was showcased for a mono-crystalline 12 nm thick UT-SOI where, in analogy with metals[34], arrays of complex nano-architectures of islands and wires (hundred nanometer high and few micrometer long, circa) were reported[28]. The key tool used to enhance the stability of the dewetting outcome against breaking leading to reproducible patterns was the creation of ad hoc dewetting fronts triggering the formation of opposite rims that move one towards the other. So far, this approach was limited to patches extending over a few μm (aspect ratio ~1/400) and, due to the anisotropy of surface energy, strictly oriented along the stable dewetting fronts (e.g. [110])[28].

Following this concept, we tested the stability against annealing of a 12 nm thick UT-SOI (at temperature between 720 and 775 °C, for periods ranging from 15 min to 2 h) patterned by electron-beam lithography and reactive ion etching in long trenches with variable pitch ($d_{LL}$ = 0.5 up to 4 μm) defining patches featuring a width $w$ ranging from ~400 nm up to ~3.8 μm. We first consider patches oriented along the stable dewetting front [110] (Figs. 1–3). Figure 1a describes the method while a more detailed description is provided in the Methods section and reference[28].

In optimized conditions (annealing temperature and time, and patterns width) we observe the formation of extremely long wires, with a length limited only by the patch design (up to 0.75 mm, Fig. 1b, c and high-resolution Supplementary Data 1). Wires with no breaks and perfectly homogeneous height and width over their full length can be formed with a 100% success rate. All the UT-SOI available in the patch ($w$ = 700 nm) collapsed in individual wires featuring a base of about 160 nm (~4 times smaller than $w$) and a height of 50 nm. In these conditions of annealing temperature and time, a simple stable dewetting front freely receding (semi-infinite UT-SOI) covers a distance of about $\Delta x$ = 650 nm (bottom-right inset in Fig. 1b), a length comparable to the overall patch width $w$.

In a similar sample, ~0.9 and ~1.9 μm wide patches ($d_{LL}$ = 1 and 2 μm, respectively) collapsed in an individual wire, whereas ~3.9 μm patches ($d_{LL}$ = 4 μm) were partially dewetted in two parallel counter-propagating rims, as revealed by AFM measurements (Fig. 2a–d). These features are attributed to a faster dewetting dynamics (and in turn to an earlier onset of the morphological instabilities) associated to smaller radius (i.e. larger curvature at the surface[35,36]) even when considering a film in contact with a substrate[37–40]: the dewetting process for the larger pitches (lower overall curvature with respect to smaller pitch), is not concluded at the end of the annealing step. It is worth noting that, although the process for $d_{LL}$ = 1 μm is faster than that one relative to $d_{LL}$ = 2 μm, also in the former case the wires did not yet break into islands (the Plateau-Rayleigh instability along the wire did not take place).

In all investigated cases, the fluctuations around the average full width at half maximum (FWHM) and heights of the wires were only a few nanometers (Fig. 2f–g) accounting for the remarkable control of the dewetting process. We also observe that for longer annealing time and higher temperature, patches of any

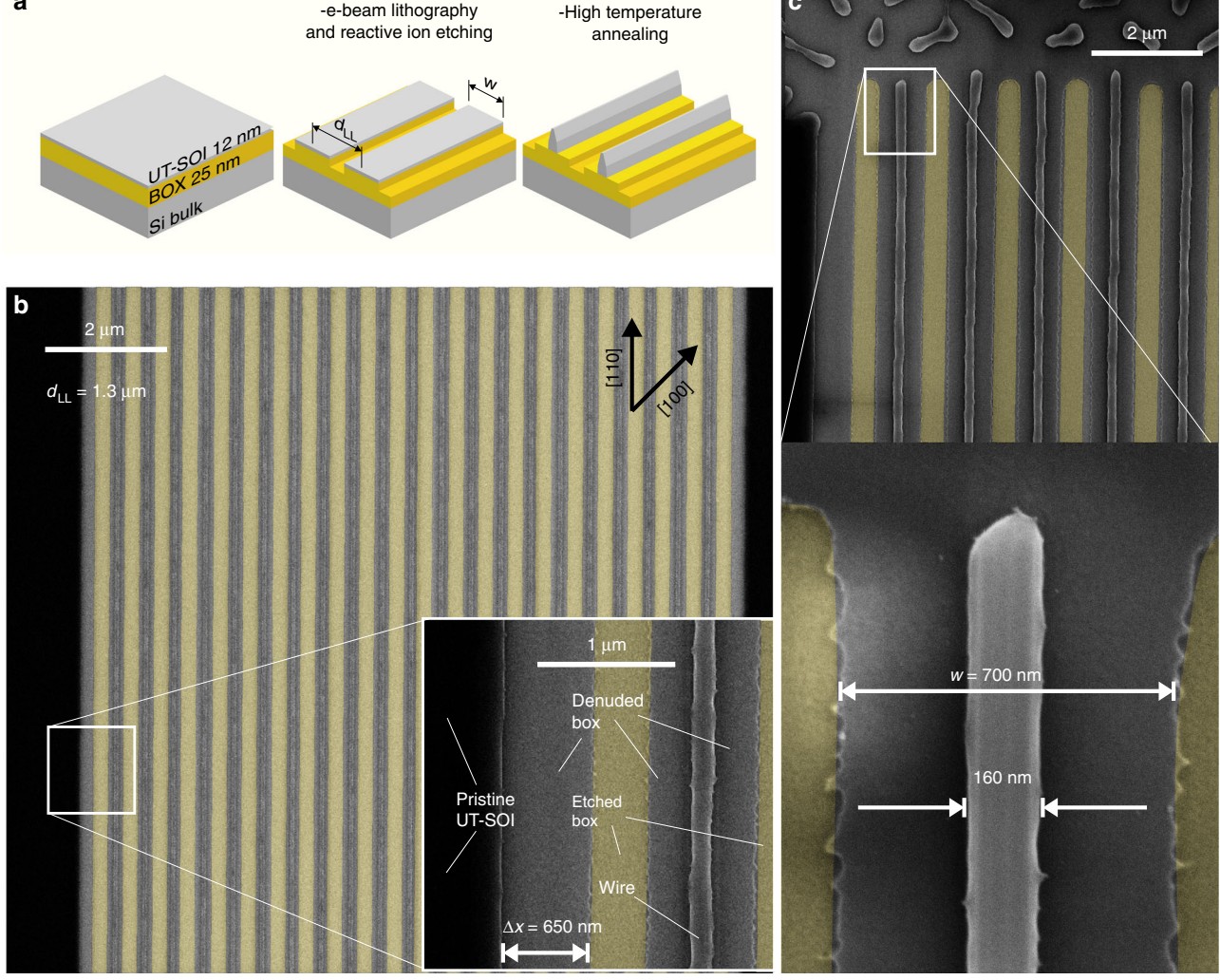

**Fig. 1 Ultra-long nano-wires formation. a** Scheme of the sample fabrication: long patches are created in a 12 nm thick ultra-thin silicon on insulator (UT-SOI) on buried oxide (BOX, SiO$_2$, 25 nm) with a pitch of $d_{LL}$ and width $w$. **b** Scanning electron microscope (SEM) image of 21 parallel nano-wires of length 0.75 mm, $d_{LL} = 1.3$ μm, and $w = 700$ nm (a high-resolution image with the full wire length is provided in the high-resolution Supplementary Data 1). Annealing temperature was 775 °C for 30′ in ultra-high vacuum (10$^{-10}$ Torr, circa). The yellow areas highlight the etched trenches. Bottom right inset: blow up of a free-propagating edge and a dewetted wire. The distance covered by a freely receding rim is highlighted by $\Delta x$. **c** Top panel: detail of the extremity of the wires. Bottom-right panel: blow-up of a wire extremity.

width produced isolated islands (eventually elongated), as expected from the conventional Plateau-Rayleigh instability (not shown)[14].

**Phase field simulations**. The former results are compared to 2D phase field simulations of surface diffusion[41,42], including surface-energy anisotropy[43,44] solved by a finite element approach[45,46] (see the Supplementary Note 1 for more details on the simulation method in use). They are performed mimicking the evolution in time of the cross-section of the experimental cases (Fig. 3, Supplementary Movies 1–6). For each investigated aspect ratio ($h/w = 1.2/100$, 1.2/200 and 1.2/400) we reproduce the dewetting dynamics, including the typical facets of the equilibrium shape of Si ({001}, {113}, {111}, and {110}) by means of the corresponding surface-energy values[47,48] (Fig. 3 a–c, left panels). We systematically compare these results with the isotropic counterpart, by averaging the energies of different orientations (Fig. 3a–c, right panels).

The relevant features emerging from this analysis are summarized as follows:

(i) For simulations reproducing $h/w = 1.2/100$ and 1.2/200 (corresponding to $d_{LL} = 1$ μm and 2 μm, respectively Fig. 3a, b, left panels), the patch effectively collapses in a single wire, directly reproducing the corresponding experiments (Fig. 3d, left and central panels), thus providing a confirmation of the diffusion-limited kinetics at play. Discrepancies between experiments and theory for $d_{LL} = 1$ μm can be attributed to convolution effects with the AFM tip leading to an overestimation of the wires FWHM also reflected in the large discrepancy between the wires width shown in Figs. 1c, 2e.

(ii) The simulations for $h/w = 1.2/400$ (corresponding to $d_{LL} = 4$ μm, Fig. 3c, left panel) show first the formation of two parallel, counter-propagating rims, leading finally to a breakup in two parallel wires as final stage (not observed in the experiment, as the dewetting process is not complete, Fig. 2c). A good agreement with experiments is found when focusing on the intermediate time steps (Fig. 3d, right panel).

(iii) Surface faceting is found to play a central role in determining a quantitative outcome of the process[49]. For $h/w = 1.2/100$ a single wire is obtained with and without

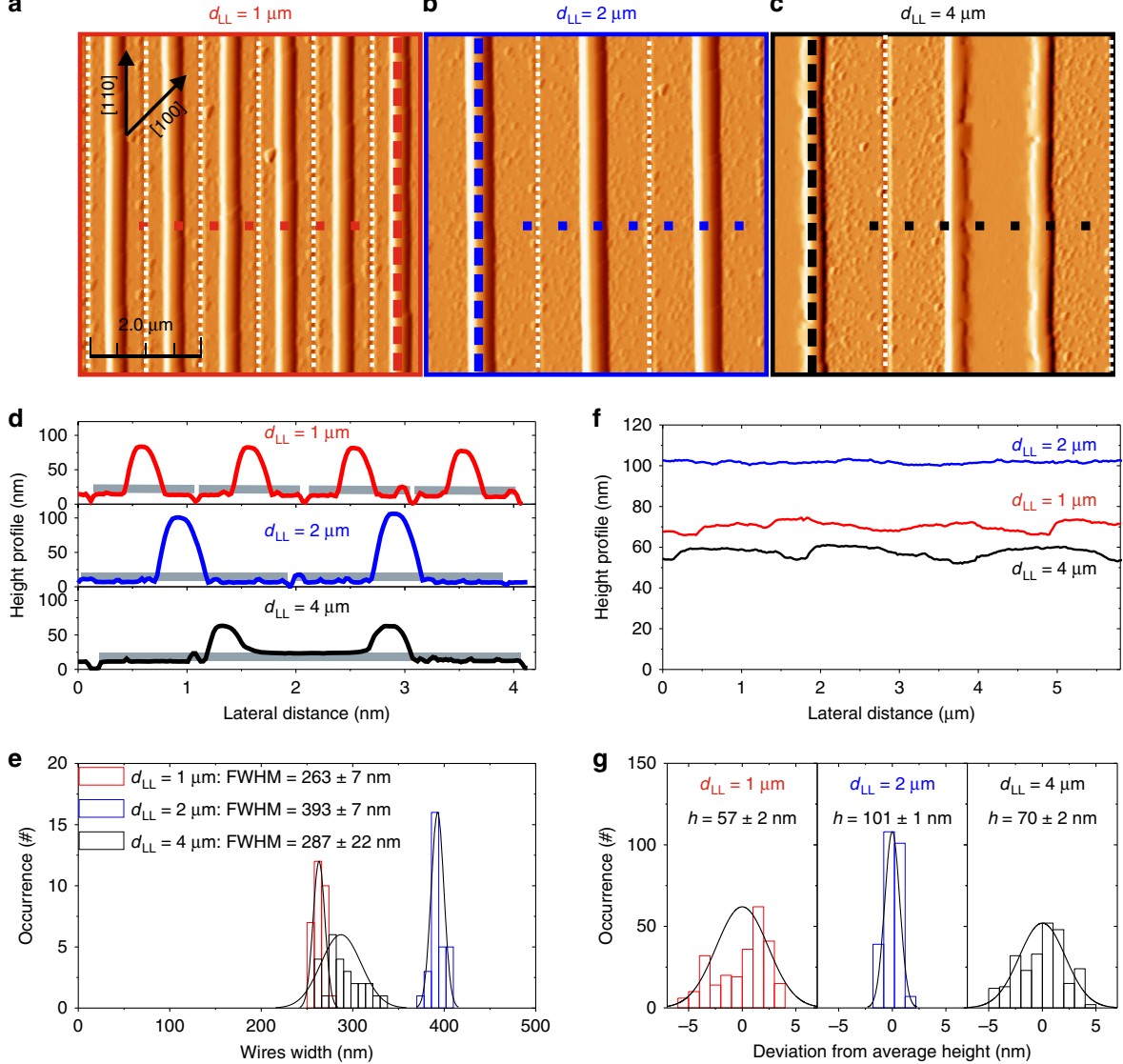

**Fig. 2 Atomic force microscopy of parallel wires oriented along the stable dewetting front. a–c** Atomic force microscopy (AFM) images of wires arrays dewetted at 720 °C for 120′ for $d_{LL}$ = 1, 2 and 4 µm respectively, and a length of 75 µm. The width of the etched trenches was about 100 nm (highlighted by vertical, white dotted lines). Horizontal dashed lines highlight the height profile shown in **d**. Vertical dashed lines highlight the sagittal height profile shown in **f**. **d** Transverse height profiles (from **a–c**, respectively top, central and bottom panel). The shaded areas highlight the original UT-SOI profile before annealing. **e** Statistical distributions of the full width at half maximum (FWHM) of the wires shown in **a–c**. The continuous lines are Gaussian envelops of the statistical distributions. **f** Sagittal profiles of the wires shown in **a–c**. **g** Statistical distributions of the wires height shown in **f** after subtraction of the height average value. The continuous lines are Gaussian envelops of the statistical distributions.

anisotropy, whereas the case $h/w = 1.2/200$ deviates from experiments, providing two parallel wires as final state of the process, when neglecting preferential orientations (Fig. 3b, right panel). Also for larger patches ($h/w = 1.2/400$, Fig. 3c, d, right panel) the isotropic case predicts three islands against the two found in the anisotropic case, confirming the tendency of the surface anisotropy forces to stabilize the patch against break-up.

(iv) For the largest patch, the experimental rims are smaller than the prediction by phase field simulations and the valleys next to the rims are not visible. This feature is attributed to an effective larger stiffness/anisotropy of the real structures with respect to those considered in these simulations. This could be readily accounted for by phase field simulations[50] at the cost of a significant increase

in the computational budget without however, delivering relevant, additional information than those discussed so far.

**Templated dewetting along arbitrary fronts.** In a different sample, similar patch arrangements are etched with a slight mis-cut of 2° circa, with respect to a stable dewetting front. We now consider patches size of 800 nm in width ($d_{LL} = 1$ µm, Fig. 4a).

The mis-cut does not impede the formation of well-ordered arrays of parallel and uniform wires, mostly intact and following the macroscopic direction imposed by the etching. Small kinks are formed during edge retraction in analogy to what was observed in the metallic counterpart[23,51–53]. However, in our case, the periodicity of the edge undulation of the Si film is not as regular as those found in metals and is linked to the presence of

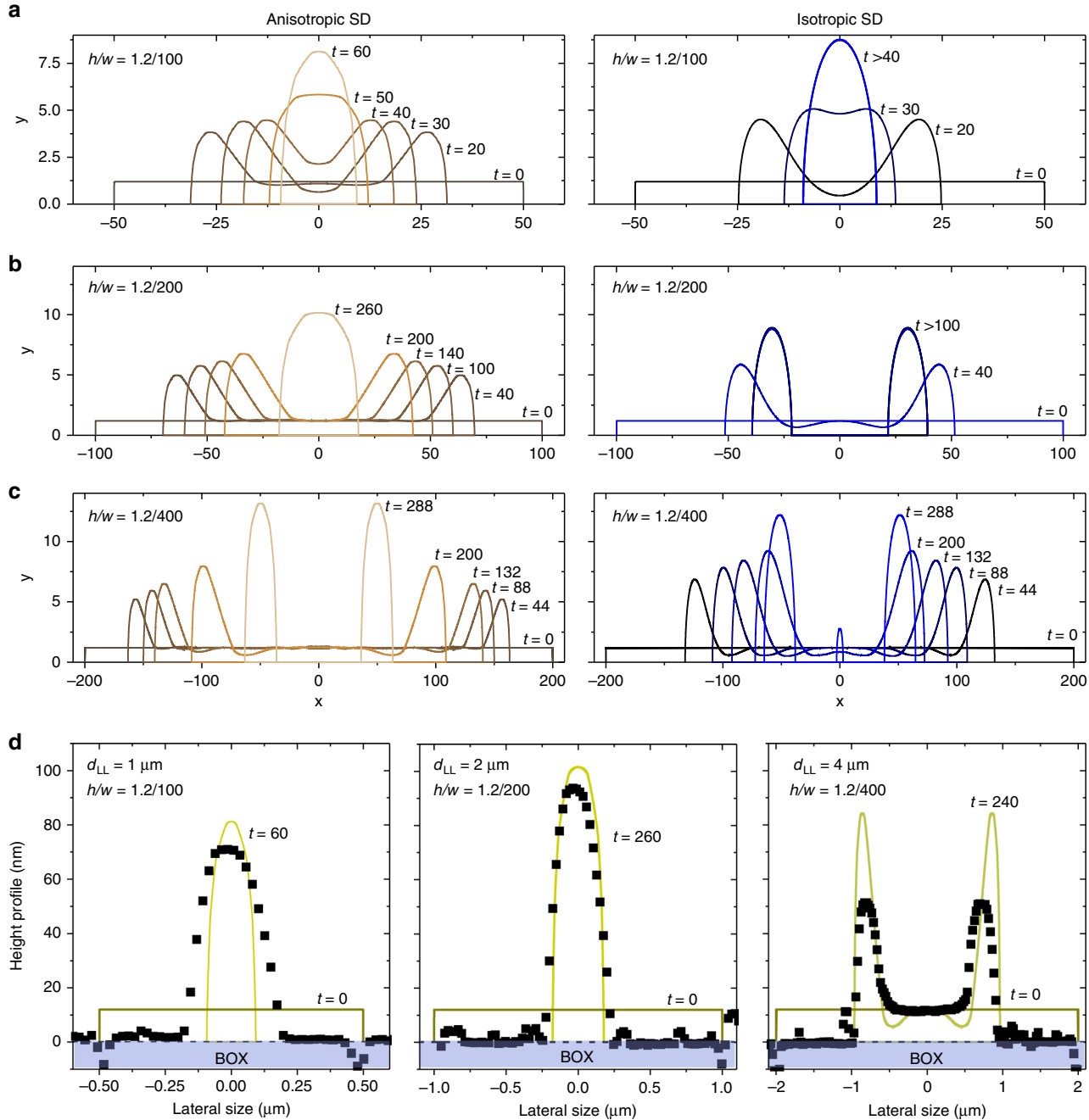

**Fig. 3 Phase field simulations of anisotropic and isotropic patches. a–c** Left panel: cross sections at different times of phase field simulations of dewetting of a patch with initial aspect ratio h/w = 1.2/100, 1.2/200, and 1.4/200, respectively, taking into account the crystal anisotropy. Right panels: same as the left panel for the isotropic case. The full dynamics is shown in the Supplementary Movies 1 and 2 for **a** left and right panels respectively; 3 and 4 for **b** left and right panels, respectively; 5 and 6 for **c** left and right panels respectively. **d** Comparison between the experimental data (symbols, from Fig. 2**d**) and phase field simulations (continuous lines). The shaded areas highlight the buried oxide (BOX) underneath the dewetted structures.

small tips at the BOX surface (highlighted by white and yellow arrows in the bottom panel of Fig. 4a) on the denuded BOX and at the sides of the wires, where they touch the BOX (these are also found in spontaneously dewetted samples and are thus not attributed to the lithographic process, see the Supplementary Note 2). All the kinks form in presence of an impurity at the wires/box interface (yellow arrows in the bottom inset of Fig. 4a) whereas in some cases only a wrinkling of the {113} and {111} facets is observed (white arrows in the bottom inset of Fig. 4a). The top {001} facet is instead always flat.

The mechanism illustrated in Fig. 4a helps in setting arbitrary orientations of the etched profile and obtaining slightly curved structures. This can be expanded to more complex patterns leading to connected networks of wires. We address this point by etching parallel patches (700 nm large and 33 μm long) and studying the effect of their orientation with respect to the crystallographic axes on the stability against breaking (Fig. 4b). Between the patches (etching highlighted by yellow areas in Fig. 4) we added several connectors with variable size and respective alignment. This design is repeated with 15° steps with

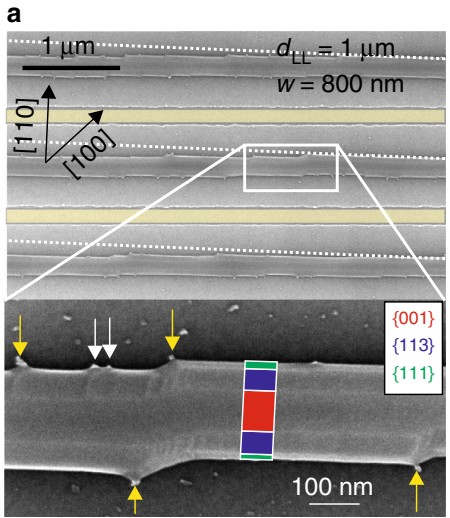

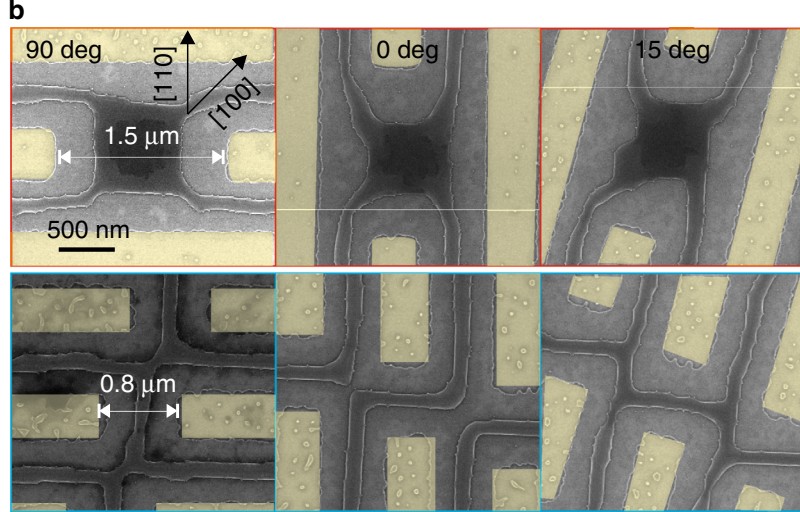

**Fig. 4 Wires tilt and connected circuits. a** SEM image of wires obtained from ∼800 nm wide patches, ∼2° mis-cut with respect to the main crystallographic axis [1-10] (highlighted by white, dashed lines) and annealed at 740 °C for 15 min. The yellow areas highlight the etched parts. Bottom inset: blow-up of a wire. White arrows highlight small tips at the wires edges; yellow arrows highlight tips in correspondence of the wire kinks. The equilibrium facets of the wire are highlighted with colored rectangles (green: {111}; blue: {113}; red: {001}). **b** SEM images of connections between parallel wires. The yellow areas highlight the etched parts. In each column, the two panels show different realizations of the connectors between two parallel wires oriented at different angles with respect to the [110] crystallographic direction. Respectively from the left to right: 90°, 0°, and 15°. The full set of images of complex circuits of three connected wires oriented at different angles with respect to the crystallographic axes is provided in the Supplementary Note 2 and high-resolution Supplementary Data 1–7 in their full 33 μm length.

respect to the [110] direction in order to cover 360° (Supplementary Note 2 and high-resolution Supplementary Data 2–7).

As found for metals[23,51] we observe a general tendency to more frequent break-up of the patches along the unstable axis [100] whereas along the stable one [110] we find one individual, elongated island as previously shown for simple wires (Supplementary Note 2 and high-resolution Supplementary Data 2). Nonetheless, for short enough annealing time, well connected structures featuring a limited number of breaks can be found in a large range of patch orientations. Up to 15° with respect to [110] the structures are not broken along the wires nor at the level of large and small connectors (respectively 1.5 μm and 0.8 μm wide, Fig. 4b). Finally, for the larger connectors the structure is robust against breaking up to 45° (high-resolution Supplementary Data 2–7).

This demonstrates that it is possible to control the continuity, connectivity and curvature of the wires up to several degrees of misalignment with respect to the stable dewetting front without any optimization. A more appropriate choice of etching design and experimental conditions (e.g. annealing time, patch width and shape as well as ad hoc additional features etched within[28]) may improve the quality of the final outcome.

**Crystalline structure of templated dewetted wires.** In order to rule out the presence of crystalline defects in the dewetted structures we performed atomic-resolution scanning transmission electron microscopy (STEM) imaging on a wire (Fig. 5). In line with previous evidences in Si- and SiGe-based islands[27,31], we observe the typical crystalline structure of bulk Si and the absence of extended dislocations. A slight crystalline disorder can be observed in some part of the wire body, at the interface with the original UT-SOI substrate (at about 12 nm from the BOX, Fig. 5c, d). This feature has been previously observed in STEM images[27,31] and we ascribe it to residual defectivity on the UT-SOI substrate, possibly due to a non ideal cleaning of the surface. For the sake of thoroughness we mention that geometric phase

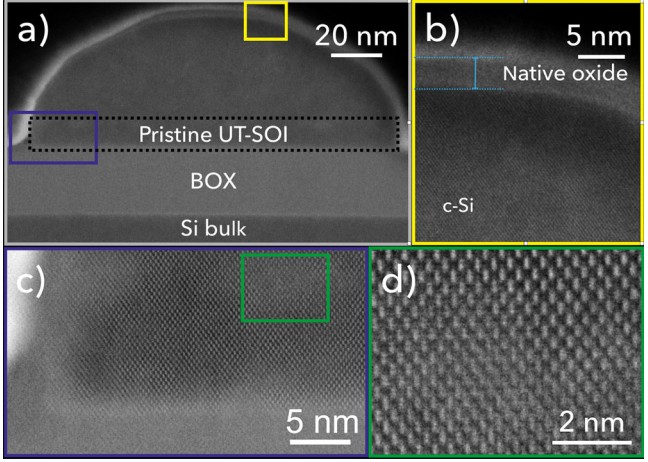

**Fig. 5 Atomic-resolution scanning transmission electron microscopy imaging. a** High Angle Annular Dark Field (HAADF) scanning transmission electron micrograph (STEM) of the section of a dewetted wire. The original UT-SOI is highlighted by a dashed rectangle. **b** Blow-up of **a** (yellow rectangle) highlighting the crystalline structure of the top part of the silicon wire (c-Si) and the native oxide surrounding it (thickness about 2.5 nm). **c** Blow-up of the bottom-left part of the wire from **a** (blue rectangle) highlighting its crystalline structure and the presence of some alloy disorder. **d** Blow up of **c** (green rectangle) highlighting the alloy disorder.

analysis performed on the full wire section does not reveal any strain in the crystal structure (not shown).

**Electric conduction from parallel wires arrays.** Through the templated dewetting process we demonstrated crystalline silicon wires formed directly on an insulating substrate. To show the potential of these structures for electronic circuits a doping procedure involving phosphorus spin-on-dopant (SOD) deposition

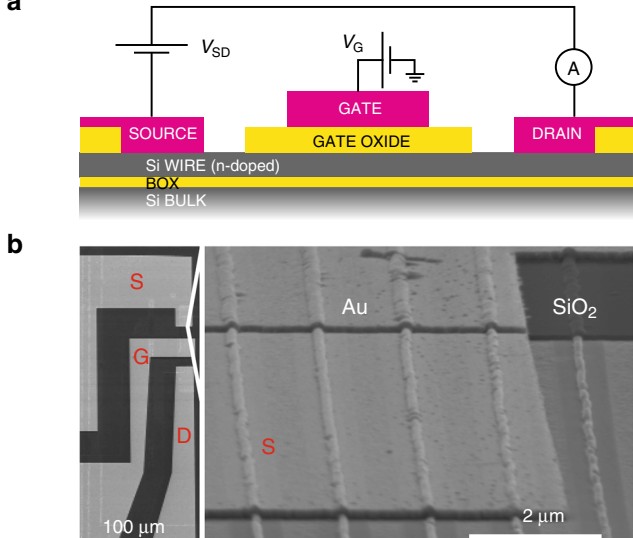

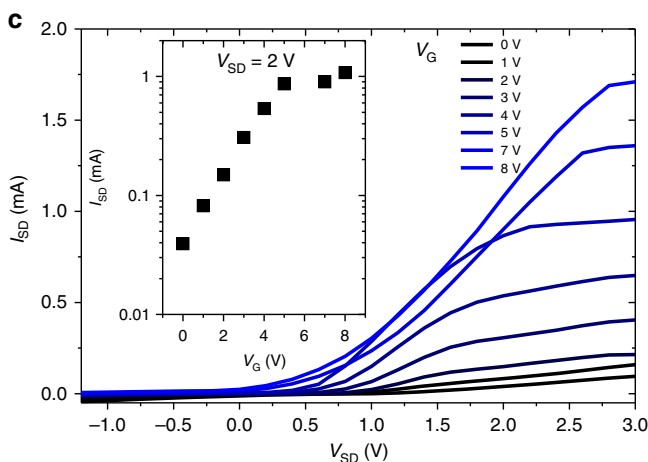

**Fig. 6 Electrical properties of parallel wires. a** Scheme of the field effect transistor (FET) wires device (doping about $3 \times 10^{18}$ cm$^{-3}$, n-type). **b** SEM images of the electric contacts (left part, source, gate and drain, respectively S, D, and G) and a detail of the S contact (right). The gold contact (Au) and the SiO$_2$ partially covering the wires are highlighted. **c** Electrical transport characteristics of 15 parallel wires (S-D current versus S-D voltage curves recorded at different G voltage. Inset: S-D current versus G voltage recorded at source-drain voltage $V_{SD} = 2$ V for 15 parallel wires.

and rapid thermal annealing treatment has been carried out on 70 µm long nanowires (base size ∼150 nm, height ∼75 nm) to render them conductive (Fig. 6 and Methods)[54]. All metal contacts (source, drain and gate, respectively S, D, and G) are 130 nm thick gold and were placed by e-beam deposition on 15 parallel nano-wires. S and D partially wet the silicon wires whereas G is separated from the wires by a 100 nm thick silicon dioxide (deposited via e-beam evaporation) covering the wires for about 30 µm (Fig. 6). The oxide leakage characteristics is about 10 pA (not shown).

S-D current curves as a function of the S-D voltage are registered for different G voltages, demonstrating a typical behavior of a field-effect transistor (FET). This transistor works in enhancement mode, where the saturation and the S-D voltage are characteristic of a n-channel FET[55]. From the I–V curves obtained on 15 parallel nanowires for different G tension, we estimate a trans-conductance ($G_{NW} = \Delta I_{SD}/\Delta V_G$, where $\Delta I_{SD}$ is

the source-drain current modulation against the corresponding change in gate tension $\Delta V_G$) of the order of ∼ µS per wire (Fig. 6).

Adopting the common approximations for nanowire-based transistors[56] and neglecting spurious effects[57], we can estimate the electron mobility with the formula $\mu_e = (L^2 G_{NW})/(V_{SD} C_{NW})$, where $L$ is the gate contact length, $V_{SD}$ is the source-drain tension and $C_{NW}$ is the gate capacitance. In a cylindrical geometry, this latter characteristic can be expressed as $C_{NW} = 2\pi\epsilon_0\epsilon_r L/\cosh^{-1}(t/R)$, with $\epsilon_0$ vacuum permittivity, $\epsilon_r$ the static dielectric constant of the gate oxide, $t_{tot} = t_{ox} + R$ distance between wire center and metallic contact, where $t_{ox}$ is the oxide thickness ad $R$ the wire radius. Assuming $\epsilon_r \sim 3.9$, $L \sim 30$ nm, $t_{ox} \sim 100$ nm, $R \sim 60$ nm (estimated as average between height and base size), we obtain a gate capacitance $C_{NW} \sim 5$ fF. With these values, we can roughly estimate an electron mobility ranging between 0.5 and $5 \times 10^3$ cm$^2$ V$^{-1}$s$^{-1}$.

## Discussion

There are several differences and advantages of our wires with respect to previous demonstrations of similar structures implemented via dewetting and other fabrication techniques, particularly regarding the simplicity and versatility of implementation, improved comprehension of the dewetting mechanism and improved quality of the structures enhancing the electrical properties of the implemented devices. In what follows we discuss all these features with respect to the existing state-of-the-art.

In this work, we extend the coherent control of dewetting by more than 2 order of magnitude, going from patches of a few µm[28] to 0.75 mm (only limited by the pattern etched prior to annealing set by the writing field of the e-beam lithography in use). This is a record aspect ratio of a ∼1/60000 patch providing uniform and reproducible nanowires. In optimal annealing conditions and patch size we obtain a 100% success in forming perfect faceted nano-wires with no breaks along their length and size fluctuations in the few per cent range.

An important difference with respect to previous reports of Si dewetting[28] is the control of patch evolution for patterns oriented along unstable fronts. The formation of kinks during dewetting (mediated by small defects at the BOX/wire interface) allows to adjust the macroscopic directions of the final structures without breaking. This feature has never been reported so far in semiconductor dewetting and it allows to curve, split and connect the wires ad libitum in order to form complex circuits.

The limit of this method is evident when considering patches larger than a few µm. Although the fluctuation of the wires width and height is always well below 10%, larger patches ($d_{LL} = 4$ µm, Fig. 2) provide wires showing a spread of their width about three times larger than those formed from smaller patches ($d_{LL} = 1$ and 2 µm, Fig. 2). This lower level of control over the rim evolution for larger structures was reported for the case of simple square patches[28]: above 3 µm, disorder effects play an important role leading to marked deviations of the dewetting dynamics with respect to what expected for ideal systems (as those shown in the simulations, Fig. 3 and reference[28]). These spurious effects could be attributed to extrinsic disorder locally perturbing the rim evolution and affecting in a more marked way larger patches with respect to smaller ones (e.g. native oxide not properly removed or other impurities present in the ultra-high vacuum used for silicon dewetting). Furthermore, small tips found at the rim/BOX interface can perturb the dewetting dynamics[28]. Thus, although these defects allow the formation of kinks along the wires and thus to curve them with respect to the preferential axes orientations, they may be a limit for a coherent control of larger patches (e.g. those that would result in two parallel wires instead of an individual one). In the present case of 12 nm thick UT-SOI, a

coherent control of the patch evolution (in absence of additional features in the initial patch design) is limited to <3 µm.

In the present work, we also managed to compare real systems with realistic models taking into account anisotropic surface diffusion. So far, simulations of templated dewetting of UT-SOI based on a phase field approach considered patches featuring an aspect ratio of, at most, 1/160[28]. This was in stark contrast with the real systems featuring a much smaller value of ∼1/400. Furthermore, this attempt to reproduce the experimental outcomes did not take into account the underlying crystal anisotropy. A reasonable agreement between experiments and simulations was found for short time evolution while showing marked discrepancies for longer times. More generally, in the last few years several theoretical works tried to tackle the anisotropic dewetting dynamic with sharp interface models for both cases of weak[58] and strong[50] anisotropy. However, in all these cases the patch aspect ratio was at most 1/60 which is pretty far from the actual experimental conditions used for metal and semiconductor dewetting. Here we used a phase field model taking into account surface diffusion and surface-energy anisotropy for a 1/1 scale case (aspect ratio up to 1/330).

Our novel theoretical understanding of the anisotropic dewetting problem allows therefore to correctly predict long-time evolution of the main features observed in experiments showing that the presence of facets (due to anisotropic surface energy) stabilizes the dewetting outcome against breaking. Isotropic models (e.g. showing two parallel wires instead of only one) fails in this task, at least for larger patches.

From a fabrication standpoint our approach offers several advantages with respect to other bottom-up methods. We implemented our wires on commercial UT-SOI wafers, available in a large set of device thickness, orientation, doping, composition (e.g. SiGe and Ge on insulator), BOX thickness, and up to 12 inches in size. These features are not matched by other methods implemented on, at most, a few inches and expensive epilayers[19–21] (e.g. for III–V-based structures).

Our structures are implemented only in two steps, etching and annealing. Thus, templated dewetting of wires offers a versatility in directions, splitting and connections not attainable with catalyst-based approaches for in-plane wires growth[59] (where all the structures were parallel) and an easier implementation with respect to recent demonstrations were the wires were bound together with complex patterning and epitaxial nano-fabrication methods[18–22,60,61] (eventually requiring cumbersome post-growth processing for the implementation of an electronic device).

Other self-assembly methods are not suitable for the formation of crystalline structures on amorphous $SiO_2$. Bottom-up methods for semiconductor-based 3D structures rely on strain-induced assembly (e.g. via Stranski Krastanov for III–V and IV–IV)[62], droplet epitaxy and droplet etching (only for III–V)[63], vapor-liquid-solid growth via gold catalyst[59], or more advanced hybrid top-down/bottom-up approaches[19]. All these strategies can be exploited only when a precise epitaxial relation holds between the substrate and the deposited material. As such, they require a crystalline support. In contrast to this, we directly produce well-ordered, monocrystalline nano-architectures on amorphous $SiO_2$ without any epitaxial relation. A straightforward consequence is that silicon dewetting provides electrical isolation of the three-dimensional nano-achitectures from the substrate, a clear advantage for the implementation of electronic devices (e.g. the field-effect transistors shown here) with respect to recent in-plane nano-architectures epitaxially grown on a III–V substrate[18,21] that becomes insulating only at very low temperature[19]. Another important difference between dewetting and recent reports of advanced epitaxial structures based on selective area growth, is

the lack of strain and alloy disorder that is known to complicate the interpretation of the device behavior[19,20].

The width of our wires is 4–5 times smaller than the initial patch width, implying that a low-resolution etching (e.g. based on optical or nano-imprint lithography) can in principle be exploited. This constitutes an advantage with respect to current hybrid top-down/bottom-up approaches, where the resolution of the lithography directly sets the final size of the structures[19,21]. In this respect, it is worth stressing that a high resolution for electron-beam lithography inevitably requires a reduced writing field, limiting the size of nanowires to rather short channel length (a few µm for both top-down[57,64] and bottom-up[18–21] approaches). In our case, a low resolution allows for mm scale structures providing an aspect ratio of the order of $1/10^4$.

Our approach allows to tune the wires height on the same substrate in contrast with top-down approaches that lead to structures with a height fixed by the thickness of the UT-SOI in use or by the etching depth. We achieved this by setting the initial patch lateral width (e.g. from $w = 0.8$ µm up to $w = 2$ µm) providing wires having a base between 150 and 400 nm and height ranging from 50 to 100 nm. In order to obtain smaller wires, it is in principle possible to use thinner UT-SOI layers (e.g. by etching a few nm of the top silicon layer).

One of the most important features of nano-wires is their electrical properties. For instance, the changes in conduction of wire-based transistors can be efficiently exploited for sensing[64] or for thermoelectricity[7] with wires having rough interfaces. Nevertheless, most works on nanowires have focused on studying and optimizing their surface smoothness. Surface defects are very common and lead to electron-surface roughness scattering influencing the charge carrier density in the underlying silicon matrix[64] modifying the electronic properties of a device (e.g. drop of the electron mobility)[65]. Thus, trans-conductance and electron mobility are important figures of merits that account for the quality of the interfaces and the limits of a wire-based device.

In our wires, the measured value of trans-conductance is $G_{NW} \sim 1$–9 µS. This is quite similar to state-of-the art FET transistors based on Si-nanowires grown via conventional bottom-up methods. In fact, for these devices, $G_{NW}$ is at most a few µS[66,67] even when considering wires having size close to those shown here[6].

From the electric characterization, a rough estimation of electron mobility in our FET transistors provides $\mu_e = 0.5$–$5 \times 10^3$ cm$^2$V$^{-1}$s$^{-1}$. These values are in line with those found in Si-based bottom-up nanowires devices[6,10,66]. For top-down FET wires (usually fabricated via e-beam lithography and reactive ion etching) typical values of $\mu_e$ lie in few hundreds of cm$^2$V$^{-1}$s$^{-1}$[56,68]. Larger values of $\mu_e$, up to ∼$10^3$ cm$^2$ V$^{-1}$s$^{-1}$, can be reached with top-down wires at the price of cumbersome fabrication methods (e.g. for smoothing the wires walls via oxidation), Ge alloying or strain engineering[69,70]. Thus, even without any kind of optimization of our devices, we can reach relatively high electron mobility. We interpret this feature as a possible consequence of the reduced surface roughness in our structures with respect to those obtained via vapor-liquid-solid growth and top-down methods.

In conclusion, we showed that dewetting, a spontaneous shape instability common to many different thin films of organic and inorganic substances, can be efficiently controlled in order to form extremely long and connected circuits of monocrystalline silicon wires on $SiO_2$. We extended the control of this process to silicon patches having the record aspect ratio of 1/60000 (to be compared with the case of metals 1/100[34], and semiconductors 1/400[28]) forming extremely elongated silicon mono-crystals.

We directly compare the experimental outcomes with 1/1 scale phase field simulations of surface diffusion that quantitatively reproduce their morphological evolution. We show a clear evidence of the key role played by faceting in stabilizing the dewetting outcome against breaking and thus for the reproducibility of the process and the stability of the final structures. Phosphorous-doped conductive Si wires are implemented showcasing the possibility to fabricate conducting nano-channels and transistors on an insulating substrate. Since the proposed approach is very general, it can be adapted to tune the Si wires aspect ratio by choosing suitable UT-SOI thickness and pattern periodicity combined with more complex, connected nano-architectures towards a full exploitation of their record length and atomically smooth facets.

Owing to a similar dewetting dynamic ruled by surface-diffusion-limited kinetics observed in SiGe alloys[32] similar results can be extended to these materials rendering this method attractive for wires formation with different materials with the perspective of band-gap engineering and carrier mobility enhancement.

## Methods

**Sample preparation**. A 12 nm-thick UT-SOI on a 25 nm thick buried oxide (BOX) was etched by electron-beam lithography and reactive etching with parallel trenches, from 0.75 mm to 70 μm long, with variable pitch (0.5–4 μm line-to-line distance, $d_{LL}$) and orientations with respect to the crystallographic axes. Thus, the samples were processed by plasma and wet chemical cleaning in $N_2$ atmosphere for 20–30 s in a 5–10% HF. Finally, they were annealed in the ultrahigh vacuum ($\sim 10^{-10}$ Torr) of a molecular beam epitaxy reactor.

**Sample doping**. Deposition of Spin on dopant (SOD, OD P508) was performed by spin-coating at 4000 rpm for 1 min and baking on a hot plate (10 min at 120 °C). Thermal diffusion of dopant was induced by rapid thermal annealing (30″ at 850 °C in $N_2$ atmosphere). The SOD was removed by wet etching for 60″ in dilute HF (1:10). The metal pads (5 nm Ti/150 nm Au), after the definition of the contact design process by electron-beam lithography, are deposited by e-beam deposition.

**Electronic imaging**. Scanning transmission electron micrograph have been acquired in z-contrast with a Cs probe corrected JEOL ARM 200F operated at 60 keV.

## Data availability

The data supporting the findings of this study are available from the corresponding authors upon reasonable request.

## Code availability

The code adopted for phase-field simulations is available upon request from M.S. (corresponding author).

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

## Acknowledgements

We acknowledge the PRCI network ULYSSES (ANR-15-CE24-0027-01) funded by the French ANR agency, the EU H2020 FET-OPEN Narciso (No. 828890) and the I-Zeb project (7784/2016) III Accordo Quadro CNR-Regione Lombardia for financial support. M.S. acknowledges the support of the Postdoctoral Research Fellowship awarded by the Alexander von Humboldt Foundation. The computational resources for phase field simulations were provided by ZIH at TU-Dresden and by the Jülich Supercomputing Center within the Project No. HDR06. STEM imaging was performed at BeyondNano CNR-IMM, which is supported by the Italian Ministry of Education and Research (MIUR) under project Beyond-Nano (PON a3 00363). We thank Dominique Chatain and Rainer Backofen for the useful discussions and Dr. Domenico Mello (STMicroelectronics, Catania) for the STEM lamella preparation. We acknowledge the Nano-TecMat platform of the IM2NP institute of Marseilles and the microscopy center of Aix-Marseille University CP2M.

## Author contributions

M.Bol., M.S and M.A. conceived the experiment. A.B., M.Bou., M.N. and I.F. performed the dewetting. M.L., S.d.C. and M.Bol. performed the lithography and the etching. L.F., J.-B.C. G.N. and A.M. performed SEM and STEM micrographs. M.S. and A.V. performed the simulations. M.Bol., M.S. and M.A. wrote and revised the paper. M.Bol., M.S., A.B., M.Bou., M.N., M.L., S.D.C., A.F., A.V., I.F., L.F., J.B.C., D.G., G.N., A.M., A.R., I.B. and M.A. commented on the paper.

## Competing interests

The authors declare no competing interests.
