## [Peer Review File · Nature Communications]

Reviewers' comments:

Reviewer #1 (Remarks to the Author):

In the manuscript Bollani et al report fabrication of ultra-long (\sim up to 0.75 mm) high aspect ratio (\sim 1/60000) monocrystalline silicon nanowires and complex inter-connected circuits through 'low' resolution etching and subsequent annealing of ultra-thin (12 nm) pre-patterned SOI wafers. The final structure of the nanowires are 5 times smaller than the defined patches and the final width depends on pre-patterned geometry. Moreover, the authors perform phase field simulations and conclude that the surface energy anisotropy is responsible for stabilizing the one-dimensional ultra-long final Si nanostructures where surface diffusion kinetics prevent these structures from breaking. Last, they demonstrate field effect transistor operation using spin-on-doping procedure onto these nanowires. Unfortunately, we cannot recommend publication of this manuscript in Nature Communications for the following reasons:

The novelty of this work relies on their earlier publication [Science Advances 3, eaao1472, 2017] where they demonstrate the control on the morphology of final Si-nanostructures on 12 nm thin SOI wafers. The key difference between these two papers is that former focuses on isotropic surface energy resulting with complex nanoarchitectures, while the current manuscript studies the anisotropic case, creating one-dimensional ultra-long Si nanowires. Moreover, the reported structures here can be easily achieved with 'normal' resolution lithography and subsequent etching without any involvement of the de-wetting process. Thus, this manuscript neither presents a new technology/technique to create one-dimensional structures nor demonstrates something novel relatively to their previously reported approach.

Although fabrication of one-dimensional semiconductor structures reported in this manuscript might be a utility for future electronic devices, it does not justify publication in Nature Communications unless something much more distinguishable was demonstrated in terms of the properties of the reported structures. In the end, the fact that standard high-resolution lithographies could be used to create similar or the same structures ultimately weakens the impact without some clear new demonstration.

Reviewer #2 (Remarks to the Author):

The manuscript describes the fabrication of parallel, silicon-based nano-wires and complex, exploiting dewetting as a tool for the fabrication process. The manuscript is original, fascinating, well-presented and scientifically sound. All the steps of the process are well described. However, some issues need clarification. Once addressed the issues listed below this reviewer expect the manuscript to be suitable for publication in Nature Communication.

In the introduction, the authors considered several top-down approaches to the fabrication of long nano-wire, but they do not consider the bottom-up approach (see for example Nat. Protoc. 7, p1668 2012). This literature or similar should be cited in the introduction.

In the abstract the authors claim the fabrication of "monocrystalline, silicon-based nano-wires" but the authors do not prove the preservation of mono-crystallinity upon the dewetting process.

The wires seem to be very homogeneous; however, Fig. 2 shows only a line profile of a few microns. Is the same everywhere? Please add a comment or a statistical analysis.

The morphology in the different side of the stripes shown in figure 2b, c seems to be different (it is difficult to say how because it seems to be a differential image). Why?

The proposed experiment is a remarkable example for nanofabrication, can the process be extended to prove other systems? Please comment on this option.

Authors reply to the referees:

Reviewer #1 (Remarks to the Author):

“[...] The novelty of this work relies on their earlier publication [Science Advances 3, eaao1472, 2017] where they demonstrate the control on the morphology of final Si-nanostructures on 12 nm thin SOI wafers. The key difference between these two papers is that former focuses on isotropic surface energy resulting with complex nanoarchitectures, while the current manuscript studies the anisotropic case, creating one-dimensional ultra-long Si nanowires.”

Although we respect the opinion of the referee, we would like to stress that:

1) in the former work (Science Advances 3, eaao1472, 2017) we showed that a coherent and precise fabrication of complex nanoarchitectures is possible only for one specific orientation of the patterns with respect to the crystallographic axes. Here we show for the first time that the fabrication can be also controlled for other crystallographic directions (never demonstrated in semiconductor dewetting of semiconductors owing to the larger stiffness of silicon with respect to metals, see for instance Ye and Thompson, Adv. Mat. 23, 1567 (2011)).

2) The size of the nanoarchitectures can be extended of more than 2 orders of magnitude in size (from 5 μm previously shown up to 0.75 mm in the present work). Nota bene: i) such a large extension cannot be simply considered as incremental; ii) such a level of control of patches with a record aspect ratio of 1:60000 has never been shown in a dewetting process, not even for the easier case of metals dewetting.

3) Also the former process (Science Advances 3, eaao1472, 2017) was anisotropic as the system in study is exactly the same (12 nm of crystalline silicon on SiO_2) and the underlying process is dewetting guided by (anisotropic) surface diffusion and *ad hoc* templates. The difference is that in the former case published in 2017, **the phase field simulations were isotropic** (not the real system) allowing just for qualitative comparisons/predictions. This provided a nice agreement with the experimental shapes, at least for short time evolution. However, the aspect ratio of the patches taken into account was far from the real ones (1/400 in experiments against 1/160 in the simulations). In this new work the simulations include, for the first time: 1) the anisotropy and facets formation allowing us to understand further and confirm the role of surface-energy anisotropy during the dewetting process and 2) a 1:1 scale model. Nota bene: we provide quantitative predictions and, to the best of our knowledge, there are not reports of a 1:1-scale benchmark of phase field simulations of anisotropic systems with real systems. This cannot be disregarded as a mere detail. It is in fact very relevant to precisely test the predictive nature of such a simulation approach. In order to put this forward we included the following sentence in the new version of the paper:

“In the last few years several theoretical works tried to tackle the anisotropic dewetting dynamic with sharp interface models for both cases of weak [74] and strong [62,75,76] anisotropy. However, in all these cases the patch aspect ratio was at most 1/60 which is pretty far from the relevant experimental conditions used for metal and semiconductor dewetting. Very recently, simulations of templated dewetting of UT-SOI based on a phase field approach considered patches featuring an aspect ratio of, at most, 1/160 [41]. This was in stark contrast with the real systems featuring a much smaller value of $\sim 1/400$. Furthermore, this attempt to reproduce the experimental outcomes did not take into account the underlying crystal anisotropy. A reasonable agreement between experiments and simulations was found for short time evolution while showing marked discrepancies for longer time. In the present case we used a phase field model taking into account surface diffusion and surface-energy anisotropy even for a 1/1 scale case (aspect ratio up to 1/330). This approach is extremely successful in reproducing the dewetting dynamics and in predicting its final outcomes including the typical facets of the equilibrium shape of silicon. This direct

comparison highlights the role of faceting in stabilizing the dewetting dynamics in contrast with isotropic systems, more prone to breaking in isolated islands.”

“Moreover, the reported structures here can be easily achieved with ‘normal’ resolution lithography and subsequent etching without any involvement of the de-wetting process. Thus, this manuscript neither presents a new technology/technique to create one-dimensional structures nor demonstrates something novel relatively to their previously reported approach.”

From the reviewer’s comments, we now understand that this point was not properly stressed in the first version of our manuscript. As he/she acknowledges in his/her report **“The final structure of the nanowires are 5 times smaller than the defined patches”**. This clearly points out that a low resolution etching can be exploited to obtain record length nanowires. This is not matched by direct top down methods. Furthermore, **‘normal’** resolution lithography to obtain such structures implies a resolution of about ~ 100 nm in patterned areas (= write field) of $1000 \times 1000 \mu\text{m}^2$, which is not actually very easy to achieve. Moreover, also high-resolution processes (e.g. e-beam lithography and reactive ion etching) are well known to produce a roughness of the structures that are in the range of ~ 10 nm.

Our method is indeed, not only a **“new technology/technique to create one-dimensional structures”** (so far solid state dewetting was never used in this context, not for semiconductors nor for metals) but it is also capable of producing record length, perfectly parallel wires featuring facets and having a shape that cannot be achieved via conventional top down methods. For instance, conventional lithography of SOI only provides parallel, vertical, lateral walls. Furthermore, the height of the etched structures obtained via top-down approaches is set either by the thickness of the initial SOI or by the depth of the etching which is not the case for dewetting where a different patch width results in a different wire height. We highlighted these aspects by adding a new sentence in the discussion:

“Using conventional top-down approaches for wires fabrication leads to structures having a height fixed by the thickness of the SOI in use or by the depth of the etching. Furthermore, in these cases, the final shape of the structures is typically limited to parallel, vertical walls and a flat top, and have a residual roughness in the ~ 10 nm range. Our approach goes beyond these limitations as it allows to obtain faceted structures (atomically smooth) with the additional option of height tuning on the same substrate. This can be achieved by setting the initial patch lateral width (e.g. $w = 1 \mu\text{m}$ provides wires having a FWHM of 263 nm and a height of 57 nm, $w = 2 \mu\text{m}$ provides wires having a FWHM of 393 nm and a height of 100 nm).”

“Although fabrication of one-dimensional semiconductor structures reported in this manuscript might be a utility for future electronic devices, it does not justify publication in Nature Communications unless something much more distinguishable was demonstrated in terms of the properties of the reported structures. In the end, the fact that standard high-resolution lithographies could be used to create similar or the same structures ultimately weakens the impact without some clear new demonstration.”

While we certainly understand the reservations of reviewer #1 we would like to stress that the main novelty of our work lies in the demonstration of several important capabilities offered by solid state dewetting and phase field modeling in the context of nano-fabrication. Although we do not deliver **“something much more distinguishable”** our demonstration offers a clear path in this direction. Our aim with this work is to showcase that silicon dewetting offers the same capabilities as the state of the art and in some aspects beyond (ease of implementation, ultra-long wires, atomically smooth surfaces, electrical isolation from the substrate), and offers a viable route to be explored for the

design of functional devices. We believe that further demonstrations based on the findings reported in this work, belong to a separate paper focused on the applications rather than the fabrication.

We would like to stress once again that, as the referee acknowledges, only “**high-resolution lithographies could be used to create similar or the same structures**” whereas our nano-wires can be obtained using standard, low resolution lithography. Furthermore, as previously remarked, the structures obtained via top-down methods are different with respect to those we show here.

Note that a quantitative and comparative assessment of the enhanced performance with respect to the systems obtained with alternative methods goes far beyond the aim of this paper. In our opinion, including in one work a record-size fabrication with an easy method, a thorough characterization and a quantitative benchmark of the obtained structures with theoretical models and a working device constitutes a complete story that deserves publication in *Nature Communications*.

Reviewer #2

“In the introduction, the authors considered several top-down approaches to the fabrication of long nano-wire, but they do not consider the bottom-up approach (see for example Nat. Protoc. 7, p1668 2012). This literature or similar should be cited in the introduction.”

We thank the referee for their remark, and accordingly to their suggestion we added that reference and a dedicated sentence in the introduction in the new version of the paper:

“On the other side, lithographic top-down methods can be used to precisely prepare a substrate for further engineering as, for instance shown for the cases of controlled wetting [19] or nano-imprint lithography [20].”

“In the abstract the authors claim the fabrication of “monocrystalline, silicon-based nano-wires” but the authors do not prove the preservation of mono-crystallinity upon the dewetting process.”

The dewetting process showcased in our work is implemented in commercial ultra-thin SOI that is a monocrystal. During the annealing process there is no possibility of amorphization, and the diffusion of atoms on the surface is guided by the crystallographic directions of the underlying silicon matrix. The presence of the typical facets of the equilibrium shape of silicon ($\{001\}$, $\{113\}$ and $\{111\}$) all along the dewetted structures *per se* readily accounts for the crystalline nature of the wires. The reason why we did not report any TEM investigation relies in the fact that this point has been largely assessed in previous reports (among the others, also by some of the authors of this work). As an example, TEM characterization of similar structures has been reported in the following papers for the monocrystalline SOI case:

- Aouassa et al., *New Journal of Physics* 14 (2012) 063038 (in Fig.4 and 8)
- Abbarchi et al., *ACS nano* 8.11 (2014): 11181 (in Fig. 1)
- Naffouti et al., *Nanotechnology* 27.30 (2016): 305602.
- Bouabdellaoui et al., *PHYSICAL REVIEW MATERIALS* 2, 035203 (2018) (in the Supplemental Information)

A similar process was studied for the dewetting of a custom-made SOI and SGOI that were amorphous in their initial condition.

- Naffouti et al., *Nanoscale*, 8(5), 2844 (2016)
- Wood et al., *ACS Photonics* 4, 873–883 (2017)

In these latter cases TEM and SEM images show the formation of polycrystalline structures after dewetting (due to the crystallization process ongoing under the action of high-temperature annealing). These islands do not present a well-defined shape nor precise facets and their shapes are quite irregular owing to the presence of several grains featuring different orientations one with respect to the other.

From the previous reports on Si and SiGe dewetting it appears that (unfortunately), complex nanoarchitectures such those shown in this work or in Science Advances 3, eaao1472, 2017, with controlled size, position and shape is only possible for the monocrystalline case. This is due to the controlled direction of the dewetting fronts ruled by the crystal directions. However, we anticipate that future work will be devoted to a better assessment of the performances of dewetting in the amorphous case, that is very relevant to practical applications and has not yet been deeply addressed.

Under the input of the referee we added the following sentence in order to clarify this point and assess the difference between the two dewetting scenarios, mono-crystal and amorphous SOI:

“Although silicon dewetting is also possible starting from an amorphous UT-SOI 43,44 , the need of a precise and controlled dewetting front in order to form regular nano-architectures, demands for mono-crystalline wafers allowing for an atom surface diffusion ruled by the underlying crystal structure. In the monocrystalline case the dewetting process only affects the shape of the layer which always remains a mono-crystals [45–48]. Differently from the case of mono-crystalline UT-SOI, previous reports of dewetting of amorphous layers led to the formation of poly-crystalline islands featuring irregular shapes and lack of precise facets. This is due to the recrystallization during annealing resulting in the formation of crystal grains featuring different orientations one with respect to the other and, for the moment, limits the use of amorphous layers to form ordered arrays of simple individual islands [43,44].”

“The wires seem to be very homogeneous; however, Fig. 2 shows only a line profile of a few microns. Is the same everywhere? Please add a comment or a statistical analysis.”

In appropriate conditions the structures are self-similar all along their length. Please note that in Fig. 3a the AFM image includes 6 wires covering an overall length of 36 μm . Within the size and shape fluctuations reported in the corresponding statistical investigation in Fig. 2, they are all identical.

The main issue we find in experiments is not related to the homogeneity of the dewetted structures (which is extremely good, as also shown in Science Advances 3, eaao1472, 201), but rather in stopping the annealing at the right moment in order to no break the structures in individual islands.

As an example, for the use of the referee, we report in the following figure the case of a non-ideal outcome of a sample, similar to that shown in the paper, where the annealing time was too long:

As shown in the statistical investigation (in d)), only one wire has no breaks along its length. All the others feature several fractures and, in general, they show some bulging, as expected in this kind of systems. This provides wires having an average length of $\sim 250 \mu\text{m}$ with fluctuations in their size. Note that a remarkable example of this kind of “modulated” wires was recently shown in Day. et al., Nature nanotechnology 10, 345 (2015). Thus, although it is not the purpose of our study, a controlled bulging is indeed of interest. Nevertheless, we choose to not discuss this “non-ideal” case that for our scope is less important.

Although an extensive AFM investigation (over 0.75 mm) is too challenging, we performed SEM images provided as Supplemental material (see S.I.Fig1.tif), showing that, in appropriate conditions of annealing temperature and time, the wire's lateral extension is perfectly constant all along their length. Note that, due to the equilibrium shape of the silicon wires featuring precise facets, the lateral size also sets the vertical one. Here, for the use of the referee we provide a high resolution image showing several wires, helping in assessing their homogeneity in different parts:

We added the following sentence in the new version of the paper:

“SEM images of the wires over they full length is provided in the Supplemental Material (S.I.Fig1). From this analysis we conclude that the dewetted structures are homogeneous in size within the fluctuations observed in the AFM investigation (Fig. 2). This result can be only obtained in the appropriate conditions of annealing time and temperature and, on the same sample, not all the patch sizes produce unbroken wires”

The morphology in the different side of the stripes shown in figure 2b, c seems to be different (it is difficult to say how because it seems to be a differential image). Why?”

In order to highlight such fluctuation we reported in Fig. 2 e the statistics of the wires width obtained from the AFM images shown in Fig. 2 a-c. It is evident that the two rims formed in 4 μm wide patches (Fig. 2 c) fluctuates 3 times more than the other two cases (of 1 and 2 μm , Fig. 2 a and b). In order to comment on this, we added the following sentence:

“Although the fluctuation of the wires width and height is well below 10%, larger patches ($d_{LL} = 4 \mu\text{m}$, Fig. 2 c) show a deviation from the average lateral extension of the two counter-propagating rims which is, in absolute values, about 3 times larger than the smaller counterparts ($d_{LL} = 1$ and $2 \mu\text{m}$, respectively Fig. 2 a and b), as highlighted in the statistics of the wires widths (Fig. 2 e). This lower control over the rim evolution for larger structures was already reported for the case of simple square patches extending over a few micrometers [41]. Above $3 \mu\text{m}$ disorder effects start to play a major role resulting in marked deviations of the dewetting dynamics with respect to what expected in an ideal system (as those shown in the simulations, Fig. 3 and reference 41). Possible explanations of this behavior relies in the presence of diffusion barriers locally perturbing the rim evolution and affecting in a more marked way larger patches with respect to smaller ones (e.g. native oxide not properly removed or other impurities present in the ultra-high vacuum used for silicon dewetting). At the same time, the small tips found at the rim/BOX interface can perturb the dewetting dynamics 41 . Thus, although these small tips allow to form kinks along the wires and then to curve them with respect to the preferential axes orientations, they constitute a limit for a coherent control of larger patches (e.g. those that would result in two parallel wires instead of an individual one). In the present case of 12 nm thick UT-SOI, a coherent control of the patch evolution (in absence of additional features in the initial patch design) is limited to less than $3 \mu\text{m}$.”

“The proposed experiment is a remarkable example for nanofabrication, can the process be extended to prove other systems? Please comment on this option.”

We are pleased by the referee’s appreciation and we follow their suggestion to explicitly mention the possibility of an extension of this method to other systems such as SiGe alloys and metals. In the new version of the paper we added the following sentence:

“Finally, owing to the common dewetting dynamic ruled by surface diffusion limited kinetics observed in other semiconductors (e.g. (Si)Ge alloys [40,48]) and metals [49], we expect that similar results can be extended to these materials rendering this method attractive for wires formation on different substrates and in different materials.”

Reviewers' comments:

Reviewer #1 (Remarks to the Author):

The authors successfully clarified some of the earlier concerns regarding the novelty of this manuscript, especially comparing with their earlier work (Science Advances 3, eaao1472, 2017). As stated in the previous report and reiterated here, we believe that the fabrication of ultra-long one-dimensional semiconductor structures reported in this manuscript can have utility for future electronic devices. Moreover, the manuscript now makes more clear how the reported method offers an easier approach for obtaining thickness control and smooth surfaces compared to conventional top-down approaches. Nevertheless, we believe the communication is incomplete in that there is not clear demonstration of unique properties (a distinguishable feature) for these structures.

With respect to this latter point, the reported FET does not make a clear case for the advantages of the small high aspect ratio nanowires. In the response letter, the electrical isolation is argued to "showcase" their technique; however, electrical isolation is expected following RIE etch and is not a major issue in current device fabrication methods. The manuscript offers an expectation of obtaining high-mobility Si-nanostructures, taking advantage of the atomically smooth surfaces. This would indeed be a clear and interesting point, but also requires quantitative measurements and comparison with conventional state-of-the-art top-down fabricated Si-nanostructures that serve as benchmarks to all in the field.

In addition, some key device fabrication/structure details are missing, including gate dielectric layer thickness, gate length, etc. For example, the inset in fig 5. shows SiO₂ partially covering the nanowires, however there is no information regarding why there is a SiO₂ layer to begin with. Is it the gate dielectric layer? If yes, what is the thickness and how is it deposited? What are the leakage characteristics?

Reviewer #2 (Remarks to the Author):

The authors addressed all my issues, now the manuscript is much better and can be accepted in the present form

Reviewer #4 (Remarks to the Author):

The authors report on the fabrication of long Si wires by a thermally induced dewetting process. it is a thorough study comparing experiments with a theoretical model.

my two major comments are in line with the comments of the other reviewers:

- with standard production lithography, and spacer technology, very narrow lines can be defined with high fidelity extending over long distances. I agree that something special should be shown in this manuscript in order to be suitable for Nature Communications. would it be possible to go beyond what is possible with current technology with the reported process. for instance, below 10-20 nm diameter ? is the electronic quality better than that of etched material? Are certain SiGe compositions possible, which are not possible by lithography/etching? could Ge/Si core/shell structures be formed by the dewetting of a uniform SiGe layer?
- the crystalline nature of the wires is not shown here. Most probably the wires are indeed not amorphous (also clear from the facets), but this dewetting process can induce crystalline defects along certain crystallographic directions, which can affect the properties. maybe this anisotropic process is different in that aspect from the isotropic one.

Erik Bakkers

Point-by-point response to the Reviewers

Reviewer #1 (Remarks to the Author):

The authors successfully clarified some of the earlier concerns regarding the novelty of this manuscript, especially comparing with their earlier work (Science Advances 3, eaao1472, 2017). As stated in the previous report and reiterated here, we believe that the fabrication of ultra-long one-dimensional semiconductor structures reported in this manuscript can have utility for future electronic devices. Moreover, the manuscript now makes more clear how the reported method offers an easier approach for obtaining thickness control and smooth surfaces compared to conventional top-down approaches. Nevertheless, we believe the communication is incomplete in that there is not clear demonstration of unique properties (a distinguishable feature) for these structures.

With respect to this latter point, the reported FET does not make a clear case for the advantages of the small high aspect ratio nanowires. In the response letter, the electrical isolation is argued to “showcase” their technique; however, electrical isolation is expected following RIE etch and is not a major issue in current device fabrication methods. The manuscript offers an expectation of obtaining high-mobility Si-nanostructures, taking advantage of the atomically smooth surfaces. This would indeed be a clear and interesting point, but also requires quantitative measurements and comparison with conventional state-of-the-art top-down fabricated Si-nanostructures that serve as benchmarks to all in the field.

AUTHORS REPLY TO REFEREE#1

We thank the reviewer for acknowledging the relevance and originality of our work. We also agree that assessing the performances of our device and compare them with the literature provides a better positioning of our results and we thank the referee for their input to our work.

By estimating trans-conductance and carrier mobility, we realized that our results are comparable to the state-of-the-art of bottom-up nanowires and are generally better than the top-down case. We ascribe this appealing feature to the presence of smooth facets and crystalline nature of our wires. Only long and cumbersome top-down fabrication methods can provide competitive results owing to an ultimate reduction in the surface roughness (other valuable alternatives to enhance the mobility are strained systems and SiGe-alloys). However, top-down structures are generally very short, owing to the ultimate resolution used for e-beam lithography, forcing the writing-field to only a few μm against ~ 1 mm long wires shown here. For these reason, we added a more detailed analysis of the electrical properties and we completely re-wrote the *Discussion* section in order to put forward the comparison with respect to the existing state-of-the-art (for both top-down and bottom-up wires).

Before detailing the replies to the referee’s comments, we would like to stress a couple of questions that we believe are important in order to understand the relevance of our work.

1) Atomically smooth facets, *per se*, are an appealing feature and a relevant distinctive property of our structures. In line with many other original fabrication processes of nanowires shown for the first time, we are providing a demonstration of the potential of our approach showing, as a proof of principle, a working device that is a *premiere* for solid state dewetting, at least to the best of our knowledge. For its ease of implementation and potential, we believe it deserves the attention of the dewetting community and, more generally of the nano-wires community. There are in fact many

reports about wire-like structures in the last 10 years following the same reasoning, even without a clear positioning with respect to previous findings or a working device:

- Tian et al, Nat. Nano. 4 824 (2009)
- Freer et al. Nat. Nano. 5, 525 (2010)
- Yao et al, Nat. Nano. 8, 329 (2013)
- Dillen, et al., Nat. Nano., 9, 116 (2014).
- Day et al., Nat. Nano. 10, 345 (2015)
- Panciera et al., Nat. Comm. 7, 12271 (2016)
- S. Gazibegovic et al., Nature 548, 434 (2017)
- Friedl et al., Nano letters 18, 2666 (2018)
- S. Vaitiekėnas, et al., Phys. Rev. Lett. 121, 147701 (2018)
- F. Krizek, et al., Phys. Rev. Mater. 2, 093401 (2018)

2) Providing an exhaustive comparison with wire-based FET devices, in general, is quite complicated, provided the plethora of devices available and all their potential applications. Depending on the use of a device, the figure of merit to optimize can be very different. As an example, for micro- and nano-electronic one would prefer a well-insulated device, insensitive to any change in the environment. On the contrary, for a sensor, one would optimize the sensitivity to surface carriers.

3) A precise estimation of carrier mobility in FET nanowires with respect to the conventional bulk counterpart is not an easy task, owing, for instance to:

- i) the approximations necessary for that estimation (Schmidt, et al. "Silicon nanowires: a review on aspects of their growth and their electrical properties." *Advanced Materials* 21.25- 26 (2009): 2681-2702);
- ii) strain induced by the top gate contact that modifies the transconductance value (Seike et al., *Applied Physics Letters* 91, 202117 (2007));
- iii) optimization of the electric contacts (CHEN, Renjie; DAYEH, Shadi A. *Metal-Semiconductor Compound Contacts to Nanowire Transistors*. In: *Nanowire Electronics*. Springer, Singapore, 2019. p. 111-158).

Thus, for these reasons, the determination of transconductance and electron mobility can be only considered as an order of magnitude.

By following the suggestion of the referee, in the new version of the *Results* and in the *Discussion* we better assess the performances of our device with respect to the present literature including top-down and bottom-up NWs transistors.

In the new version of the paper we changed the dedicated figure 6 and added these paragraphs in the section *Results*:

“Electric conduction from parallel wires arrays. Through the templated dewetting process we demonstrated crystalline silicon wires formed directly on an insulating substrate. To show the applicability of these structures to electronic circuits, a doping procedure involving phosphorus spin-on-dopant (SOD) deposition and rapid thermal annealing treatment has been carried out on 70 μm long nanowires (base size ~150 nm, height ~75 nm) to render them conductive (Fig. 6 and Methods) 11,17,72,73 . All metal contacts (source, drain and gate, respectively S, D and G) are 130 nm thick gold and were placed by e-beam deposition on 15 parallel nano-wires. S and D partially

wet the silicon wires whereas G is separated from the wires by a 100 nm thick silicon dioxide (deposited via e -beam evaporation) covering the wires for about 30 μm (Fig. 6). The oxide leakage characteristics is about ten pA (not shown).

S - D current curves as a function of the S - D voltage are registered for different gate voltages, demonstrating a typical behavior of a field-effect transistor (Fig. 6). This transistor works in enhancement mode, where the saturation and the S - D voltage are characteristic of a n -channel FET 74. From the I - V curves obtained on 15 parallel nanowires for different gate tension, we estimate a trans-conductance ($G_{\text{NW}} = \Delta I_{\text{SD}} / \Delta V_G$, where ΔI_{SD} is the source-drain current modulation against the corresponding change in gate tension ΔV_G) of the order of $\sim \mu\text{S}$ per wire (Figure 6). Adopting the common approximations for nanowire-based transistors 7,75 and neglecting spurious effects 76, we can estimate the electron mobility with the formula $\mu_e = (L^2 G_{\text{NW}}) / (V_{\text{SD}} C_{\text{NW}})$, where L is the gate contact length, V_{SD} is the source-drain tension and C_{NW} is the gate capacitance. In a cylindrical geometry, this latter characteristic can be expressed as $C_{\text{NW}} = 2\pi\epsilon_0\epsilon_r L / \cosh^{-1}(t/R)$, with ϵ_0 vacuum permittivity, ϵ_r the static dielectric constant of the gate oxide, $t = t_{\text{ox}} + R$ distance between wire center and metallic contact, where t_{ox} is the oxide thickness and R the wire radius. Assuming $\epsilon_r \sim 3.9$, $L \sim 30$ nm, $t_{\text{ox}} \sim 100$ nm, $R \sim 60$ nm (estimated as average between height and base size), we obtain a gate capacitance $C_{\text{NW}} \sim 5$ fF. With these values we can roughly estimate an electron mobility ranging between 0.5 and 5 10^3 $\text{cm}^2/(\text{Vsec})$.”

and in the section *Discussion*, where we added the following comment about the measured transconductance and mobility values:

“One of the most important features of nanowires is their electrical properties. For instance, the changes in conduction of wire-based transistors can be efficiently exploited for sensing 90,92,95–97 or for thermoelectricity 9 with wires having rough interfaces. Nevertheless, most works on nanowires have been devoted in studying and optimizing their surface smoothness. In fact, one of the intrinsic features of most nanowire devices (that are systems having large surface-to-volume ratios as the wire cross-section decreases), is the presence of surface defects. These lead to electron-surface roughness scattering influencing the charge carrier density in the underlying silicon matrix 92,98,99 modifying the electronic properties of a device (e.g. drop of the electron mobility) 92,100–106. Thus, trans-conductance and electron mobility are important figures of merits that account for the quality of the interfaces and the limits of a wire-based device applicability.

In our wires the measured value of trans-conductance is $G_{\text{NW}} \sim 1$ -9 μS . This is quite similar to what found in state-of-the art FET transistors based on Si-nanowires grown via conventional bottom-up methods. In fact, for these devices, G_{NW} is at most a few μS 39,107–117 even when considering wires having size close to those shown here 6. SiGe-based core-shell structures can eventually provide larger values of G_{NW} especially at low temperature 118. Similar considerations hold for the top-down counterpart, where strain-engineering can provide G_{NW} up to 4 μS 75,87 whereas larger values can be attained via SiGe-based hetero-structures 89,119.

From the electric characterization, a rough estimation of electron mobility in our FET transistors provides $\mu_e = 0.5$ -5 10^3 cm^2/Vs . These values are in line with those found in Si-based bottom-up nanowires devices 6,12,107,120–125. For top-down FET wires (usually fabricated via e -beam lithography and reactive ion etching) typical values of μ_e lie in few hundreds of cm^2/Vs 75,86,88,93,132,133. Larger values of μ_e , up to $\sim 10^3$ cm^2/Vs , can be reached with top-down wires at the price of cumbersome fabrication methods (e.g. for smoothing the wires walls via oxidation), Ge alloying or strain engineering 132,134,135. Thus, even without any kind of optimization of our devices, we can reach relatively high electron mobility. We interpret this feature as a possible

consequence of the lack of surface roughness present in both VLS grown nanowires and top-down methods.”

For the use of the reviewers we are reporting here some values of electron mobility in FET transistors extracted from the literature. For each paper we report the value of μ_e , the fabrication method (e.g. top-down, bottom up, VLS) and some relevant information concerning the features of the wires in use. Where not explicitly specified, the material of the wires is silicon. In the following list we also report the cases of SiGe-based core-shell and III-V nanowires transistors that are not included in the paper (we prefer to compare with Si-based transistors only).

Electron mobility (units $\text{cm}^2/(\text{Vs})$) for different systems:

INTRINSIC SILICON

>1350 $\text{cm}^2/(\text{Vs})$

BULK TRANSISTOR

-Neamen DA (2003) Semiconductor Physics and Devices (McGraw–Hill, New York), p 161

>1000 $\text{cm}^2/(\text{Vs})$

TOP-DOWN WIRES

-Singh IEEE TRANS. ELECT. DEV 55, 2008 3107 (REVIEW)

-Granzner IEEE TRAN ELECT DEV 61, 2014 3601 (ROUGHNESS STUDY)

-Hashemi, Pouya (2010). Gate-all-around silicon nanowire MOSFETs: top-down fabrication and transport enhancement techniques (Doctoral dissertation, Massachusetts Institute of Technology).

-Ryu et al. Nanotechnology 24 (2013) 315205

200 $\text{cm}^2/(\text{Vs})$

-Suk et al IEDM Tech. Dig., 2007,552

200

-Tezuka IEDM Tech. Dig., 2007, 887

600

-Wangg Nano Lett. 2006, 6, 1096

100

-Gunawan Nano Lett. 2008, 8, 1566

370

-T. Tezuka, IEDM Tech. Dig., 2007, 887

600

-Tezuka, T., *IEEE Int. Elec. Dev. Meeting*. IEEE, 2007. thermal etching

900 (Strained SOI, SGOI+)

-Moselund IEEE TRAN ELECT DEV 57, 2010 866 thermal ox)

1000 (Strained bent Si +

-Chen IEEE ELECT. DEV. LETT 30, 2009 1203

500

-Lee IEEE TRAN NANOT. 11, 2012 565

1000 (110 direction channel)

-Diss. Massachusetts Institute of Technology, 2010.

600 (Strained SOI)

-Sato *Solid-state electronics* 65 (2011): 2-8.

700 (angle engineering)

-Zeng *IEEE Trans. Elect. Dev.* 64 2485

100-2000 (THEORETICAL,

ROUGHNESS STUDY)

BOTTOM-UP WIRES

Lu IEEE TRANS. EL. DEV., 55, 2008 2859 (REVIEW)

Lu J. Phys. D: Appl. Phys. 39 (2006) R387 (REVIEW)

Schmidt Adv. Mater. 2009, 21, 2681 (REVIEW)

-Cui, Nano Lett., 3, 149, 2003

2000/1350 $\text{cm}^2/(\text{Vs})$ (VLS)

-He, Nature nanotechnology , 1(1), 42 2006.	30 (VLS)
-Huang, Science 294.5545 (2001): 1313	200 (VLS)
-Chen, Nano letters , 2015, 16, 420	255 (VLS, Ge/Si Core/Shell)
-Nguyen Nano letters , 2014, 14, 585	300 (VLS, Ge/Si Core/Shell)
-Dillen et al., , Nature nanotechnology , 9(2), 116	1800 (VLS GeSi core/shell 77 K)
-Xiang, Nature 441.7092 (2006): 489	640 (SiGe Core/Shell)
-He, (2018) Phys. Chem. Chem. Phys 20, 3888	800 (SiGe theoretical)
-Neophytou Phys. Rev. B , 2011, 84 , 085313.	2000 (SiGe theoretical)
-Tomioka Nature 2012, 488 (7410), 189	1170 (VLS InGaAs)
-Huang, Pure Appl. Chem. , 76, 2051 2004	4000 (VLS, InP)

-Dayeh, *Small*, 3 326, 2007; Thelander, *Sol. St. Com.*131, 573, 2004

	3000 (VLS, InAs).
-Xiang Nano Lett. , 6, 1468, 2006	21000 (VLS, GaN/AlN/AlGaIn 5K)
-Jiang, Nano Lett. , 7, 3214 2007	18000 (VLS, InAs/InP, <100K)
-Zheng Adv. Mater. 2004, 16 1890	70 (ZnO)
-Krizek Physical Review Materials 2, 093401 (2018)	7600 (InAs Low-T)
-Tsivion Science 2011 100	250 (GaN)
-Fan Adv. Mater. , 2009, 21, 3730	2000 (InAs)

In the response letter, the electrical isolation is argued to “showcase” their technique; however, electrical isolation is expected following RIE etch and is not a major issue in current device fabrication methods.

We agree with the referee that the electrical isolation *per se* does not demonstrate an advantage of our technique, at least with respect to top-down methods where the BOX separates the wires from the substrate. Nevertheless, it is an important distinctive feature of silicon dewetting and we believe that, at least in this respect, comparing with top-down approaches (e.g. e-beam and RIE) is misleading. In fact, our discussion of this point was devoted to compare solid state dewetting with other bottom-up self-assembly methods for 3D structures that typically cannot be performed on an (amorphous) insulating support (as SiO₂).

Note for instance that for advanced epitaxial structures (e.g. selective area growth on III-V), all the devices are in contact with the substrate (F. Krizek, et al., *Phys. Rev. Mater.* 2, 093401 (2018)). This substrate is a semiconductor and becomes insulating only at very low temperature. In other cases, very complicate nanowire transfer and contacts deposition processing are necessary in order to obtain a final device (S. Gazibegovic et al., *Nature* 548, 434 (2017)). In our case this is not necessary, as all the devices are well-separated from the substrate by a thermal oxide. An additional advantage of dewetting is that it does not rely on strain and thus is less prone to intrinsic alloy disorder in contrast with III-V compounds

In order to better stress these point we changed the text in the *Discussion*:

“Solid state dewetting allows the formation of wires on amorphous SiO₂ in contrast to most self-assembly methods. Bottom-up methods for semiconductor-based 3D structures rely on strain-induced assembly (e.g. via Stranski Krastanov for III-V and IV-IV) 83, Droplet epitaxy and droplet etching (only for III-V) 84, vapor-liquid-solid growth via gold catalyst 80, or more advanced hybrid top-down/bottom-up approaches 24,26. All these strategies can be exploited only when a precise

epitaxial relation holds between the substrate and the deposited material. As such, they demand for a crystalline support. In contrast to this, we directly produce well-ordered, monocrystalline nano-architectures on amorphous SiO₂ without any epitaxial relation. A straightforward consequence is that silicon dewetting provides electrical isolation of the 3D nano-achitectures from the substrate, a clear advantage for the implementation of electronic devices (e.g. the field-effect transistors shown here) with respect to recent in-plane nano-architectures epitaxially grown on a III-V substrate 25,28 that becomes insulating only at very low temperature 26. Another important difference between dewetting and recent reports of advanced epitaxial structures based on selective area growth, is the lack of strain and alloy disorder that is known to complicate the interpretation of the device behavior 26,27. ”

%%

Reviewer #2 (Remarks to the Author):

The authors addressed all my issues, now the manuscript is much better and can be accepted in the present form.

AUTHORS REPLY TO REFEREE#2

We kindly thank Reviewer#2 for his/her report and support.

%%

Reviewer #4 (Remarks to the Author):

The authors report on the fabrication of long Si wires by a thermally induced dewetting process. It is a thorough study comparing experiments with a theoretical model. My two major comments are in line with the comments of the other reviewers:

Erik Bakkers

AUTHORS REPLY TO REFEREE#4

We thank the referee for comments and suggestions. We understand his concerns shared with reviewer#1. In the new version of the paper we address the comments and we rewrote the *Discussion* section in the light of a more detailed analysis of the electrical properties of the FET transistor. Please see the reply to reviewer#1 for comments concerning the electrical conduction properties of our wires.

“- with standard production lithography, and spacer technology, very narrow lines can be defined with high fidelity extending over long distances. ”

Although some top-down method can provide narrow lines over long distances (e.g. by deep UV interferential lithography) the processing to get such an extreme resolution requires very long processes, involving expensive and polluting steps (see for instance HOBBS, Richard et al., Semiconductor nanowire fabrication by bottom-up and top-down paradigms. *Chemistry of*

Materials, 2012, 24.11: 1975-1991). Besides, all the wires are parallel, with no curves nor splitting. Roughness of the sidewalls is in any case determined by the etching method.

In case of free patterns etching (when interferential lithography cannot be used) involving e-beam lithography, reaching high resolution over mm scales as we do here, is problematic, as the sharp writing-beam for obtaining narrow lines obliges to use a small field of view, allowing only for very short nanowires. Here, we give a demonstration that, even with a relatively low resolution in etching, narrow structures can be obtained at the end of a simple annealing process. In addition to this, we can obtain objects that are monocrystals and atomically smooth over their full length that is of the order of 1 mm.

These latter details are important for systems featuring a large surface-per-volume ratio as it is expected to improve the electron mobility. Note also that, for top-down wire devices, in order to recover good electrical performances, additional annealing steps are employed for reducing surface roughness and dangling bonds. All these post-fabrication steps are not necessary in our case and this, *per se* represents an important and distinctive feature of our method along with several other important aspects that have been discussed in the paper. See the reply to reviewer#1 for a detailed discussion of this point.

“I agree that something special should be shown in this manuscript in order to be suitable for Nature Communications.”

In the previous version of the paper we highlighted more fundamental aspects of the novelty of our work that we believe are relevant to several communities and that were never shown so far: control of dewetting over record scales in silicon; a 1:1 comparison of experiments with anisotropic phase field simulations providing a quantitative agreement and addressing the role of faceting and stiffness; the possibility to control dewetting in order to obtain connected and ordered structures along unstable dewetting fronts; structures that are 5 times sharper with respect to the initial etching; a working electronic device from a dewetted structure that is a *premiere* for solid state dewetting.

By following the input of the referees, in the new version of the paper we improved the discussion of the FET transistor. In particular, as also required by reviewer#1, we try to assess the role of the facets of our wires (that is indeed a relevant aspect) with respect to common top-down fabrication by RIE that inevitably produce some surface defects in the sidewalls of the etched wires. See also the reply to reviewer#1 for more details.

“would it be possible to go beyond what is possible with current technology with the reported process. for instance, below 10-20 nm diameter ? ”

It is in principle possible to tune the size of the final structures in 3 different ways:

- 1) reducing the initial thickness of the SOI,
- 2) by producing a large wire and thinning it via oxidation of the external layers and
- 3) for a fixed initial SOI thickness, etching patches with a smaller lateral width (as shown in Figure 2 in the paper).

We added the following sentences to comment on this point:

“Our approach allows to tune the wires height on the same substrate in contrast with top-down approaches that lead to structures with a height fixed by the thickness of the SOI in use or by the etching depth. This can be achieved by setting the initial patch lateral width (e.g. from $w = 0.8 \mu\text{m}$ up to $w = 2 \mu\text{m}$) providing wires having a base between 150 and 400 nm and height ranging from 50 to 100 nm. In order to obtain smaller wires, it is in principle possible to use thinner UT-SOI

layers (e.g. by etching a few nm of the top silicon layer). Exploiting this idea, Si and Ge islands with controlled size (from few nm up to hundreds of nm) were obtained by choosing different UT-SOI and UT-GOI thickness 94 .

“is the electronic quality better than that of etched material? ”

Generally speaking yes, although cumbersome post-etching processing can reduce the sidewalls roughness of top-down wires and recover a comparably high electron mobility. See the reply to reviewer #1 and the new version of the *Discussion*.

Are certain SiGe compositions possible, which are not possible by lithography/etching? could Ge/Si core/shell structures be formed by the dewetting of a uniform SiGe layer?

It is indeed possible and important to transfer this technique to SiGe compounds, and this is exactly what we are studying since a few years ago. We recently produced several SiGe-based structures as shown in our papers:

- M Bouabdellaoui et al, Phys. Rev. Mat. 2, 035203 (2018)
- Wood et al, ACS photonics 4, 873 (2017)
- Naffuti et al., Nanotechnology 27, 305602 (2016)
- Naffouti et al., Small 12 , 6115 (2016)

We also have preliminary results where we produced complex nano-architectures, wires and rings either by supplying Ge during dewetting or directly dewetting a SiGe-based crystalline layer on BOX. Moreover, further engineering of the composition profile is possible by selective oxidation of Si thus enriching the structures with Ge and playing with strain (see for instance David et al., Nano letters 17 (12), 7299 (2017), Naffouti et al., Nanotechnology 27 (2016) 305602).

We addressed this point in the conclusions when providing some possible perspective to our work:

“Owing to a similar dewetting dynamic ruled by surface diffusion limited kinetics observed in SiGe alloys 50 similar results can be extended to these materials rendering this method attractive for wires formation with different materials with the perspective of band-gap engineering and carrier mobility enhancement.”

“- the crystalline nature of the wires is not shown here. Most probably the wires are indeed not amorphous (also clear from the facets), but this dewetting process can induce crystalline defects along certain crystallographic directions, which can affect the properties.”

Solid state dewetting of a mono-crystalline layer transforms a flat mono-crystal in a 3D mono-crystal. This proceeds by moving atoms from a site of the crystal matrix of the solid (e.g. the edges of the film where the chemical potential is larger) to another crystal site (with lower chemical potential). Thus, there are not structural defects nor disorder (the only source of disorder may be a non ideal surface cleaning before dewetting) and this is why we did not show any TEM characterization in the previous version of the paper.

The issue of the crystalline structure of dewetted structures has been already assessed very clearly in many works in the last 30 years. Just to mention a few, see for instance some reports where TEM

analysis accounts for the crystalline state of the dewetted structures obtained in a plethora of different conditions:

- Abbarchi et al., ACS Nano 8, 11181 (2014);
- Naffouti et al. Nanotechnology 27, 305602 (2016);
- Naffouti et al., Nanoscale 8, 7768 (2016);
- Mitsai et al. Nanoscale (2019) DOI: 10.1039/c9nr01837a;
- Korzec et al., JAP. 115, 074304 (2014);
- Aouassa et al., New Journal of Physics 14, 063038 (2012),
- Burhanudin et al., Thin Solid Films 508, 235 (2006);
- Shklyaev et al., Physics Uspekhi 51, 133 (2008);
- Kononchuk Solid State Phenomena 156, 69 (2010);
- Rocsen et al., Appl. Phys. A, 108, 719 (2012);
- Sudoh et al., C. R.Physique 14, 601 (2013);
- Sudoh et al, JAP 108, 083520 (2010);
- Sutter et al., APL 88, 141924 (2006);
- Sutter et al., APL 85, 15 , 3148 (2004);
- Sutter et al., Nanotechnology 17, 3724 (2006)

As required by the referee, in order to assess the crystalline nature of the dewetted structures, we performed TEM analysis on wires obtained via templated dewetting. In line with the other reports on similar systems, we provide the evidence of a perfect crystalline state of the wires. This is not surprising and we added this information in main text with a dedicated figure and corresponding comments.

In the section Results we added a dedicated sub-section “*Crystalline structure of templated dewetted wires.*” and this comment to the corresponding figure:

“Crystalline structure of templated dewetted wires. In order to rule out the presence of crystalline defects in the dewetted structures we performed atomic-resolution STEM imaging on a wire (Fig. 5). In line with previous evidences in Si- and SiGe-based islands 43,48,49,71, we observe the typical crystalline structure of bulk Si and the absence of extended dislocations. A slight crystalline disorder can be observed in some part of the wire body, at the interface with the original UT-SOI substrate (at about 12 nm from the BOX, Fig. 5 c) and d)). This feature has been previously observed in STEM images 43,48,49,71 and we ascribe it to residual defectivity on the UT-SOI substrate, possibly due to a non ideal cleaning of the surface. For the sake of thoroughness we mention that geometric phase analysis performed on the full wire section does not reveal any strain in the crystal structure (not shown).”

Further comments about the implications of a crystalline structure and smooth interfaces is provided in the the section *Discussion*.

“maybe this anisotropic process is different in that aspect from the isotropic one.”

Although we are not sure to what the reviewer refers exactly when he mentions the isotropic process, we stress that any diffusive process of surface atoms on a crystal is intrinsically anisotropic (surface atoms diffusion always depends on the crystallographic directions). We discussed the isotropic process only in the simulations.

Note that, including surface diffusion with strong-anisotropy of the surface energy, allowing for a correct description of the dynamic of dewetting including facets, is a very demanding computational task. When dealing with continuum modeling, required here by the size of the structures we aim to

describe, this would require the integration of high-order partial differential equation (up to the 6th order). Moreover, this should be in principle combined with advanced numerical-integration methods in terms of spatial and time discretization/adaptivity, in particular if the aim is to fully describe dewetting process in 3D. This is why in our previous work (*Naffouti et al., Science advances 3, eaao1472 (2017)*) we tackled the simulation of surface diffusion with isotropic surface energy only, combined with the fact that this approach was able to assess the main mechanism at play. Indeed, in that case the early stages of dewetting experiments were correctly reproduced. A discrepancy of 10-20% between simulations and experiments (in terms of deviations with respect to the shape obtained) was found for long time evolution. Still, the isotropic simulations were correct in describing the overall features of complex nano-architectures evolving under an anisotropic process.

In the present work, exploiting the symmetry of the system, which allows to study the evolution of stripes by focusing on their cross sections by 2D simulations, we go beyond a simplified picture by taking into account also anisotropic surface energy. This is actually necessary to correctly describe the long-time evolution of the system and provide quantitative predictions: for 2 μm wide stripes the prediction of isotropic models is two wires, whereas the real systems show only one wire. This latter feature is correctly predicted by including anisotropy. In order to better clarify these points we changed the text with this new paragraph:

“In the present work we managed to compare real systems with realistic models taking into account anisotropic surface diffusion. So far, simulations of templated dewetting of UT-SOI based on a phase field approach considered patches featuring an aspect ratio of, at most, 1/160 44. This was in stark contrast with the real systems featuring a much smaller value of $\sim 1/400$. Furthermore, this attempt to reproduce the experimental outcomes did not take into account the underlying crystal anisotropy. A reasonable agreement between experiments and simulations was found for short time evolution while showing marked discrepancies for longer time. More generally, in the last few years several theoretical works tried to tackle the anisotropic dewetting dynamic with sharp interface models for both cases of weak 77 and strong 65,78,79 anisotropy. However, in all these cases the patch aspect ratio was at most 1/60 which is pretty far from the actual experimental conditions used for metal and semiconductor dewetting. Here we used a phase field model taking into account surface diffusion and surface-energy anisotropy for a 1/1 scale case (aspect ratio up to 1/330).

Our novel theoretical understanding of the anisotropic dewetting problem allows therefore to correctly predict long-time evolution of the main features observed in experiments showing that the presence of facets (due to anisotropic surface energy) stabilizes the dewetting outcome against breaking. Isotropic models (e.g. showing two parallel wires instead of only one) fails in this task, at least for larger patches.”

REVIEWERS' COMMENTS:

Reviewer #4 (Remarks to the Author):

all my remarks have been adequately addressed. I think it is a very nice paper, which i would recommend to publish in Nature Communications.

Erik Bakkers

Reviewer #5 (Remarks to the Author):

The authors successfully demonstrate smooth and long Si wires through a relatively easy thermally induced dewetting process. The study is comprehensive and the paper was well written. The authors sufficiently discuss the advantages of their approach compared to that of top-down and bottom-up approach, namely the higher quality surfaces of the wires and the length of the wires, that could be achieved with relative ease. However, the authors still fail to demonstrate their comparative advantages – in what technology/devices that the two advantages are simultaneously needed to substantially improve the efficiency/performance of the devices. While it is ideal for the authors to have this demonstration in the paper, I am of the opinion that even without the demonstration, the paper has enough new results to be published in Nature Comm.

We thank both reviewers for their positive assessment of our work. In what follows we provide a point-by-point answer to their concerns.

Reviewer #4 (Remarks to the Author):

all my remarks have been adequately addressed. I think it is a very nice paper, which i would recommend to publish in Nature Communications.

Erik Bakkers

Authors reply.

We kindly thank the referee.

Reviewer #5 (Remarks to the Author):

The authors successfully demonstrate smooth and long Si wires through a relatively easy thermally induced dewetting process. The study is comprehensive and the paper was well written. The authors sufficiently discuss the advantages of their approach compared to that of top-down and bottom-up approach, namely the higher quality surfaces of the wires and the length of the wires, that could be achieved with relative ease. However, the authors still fail to demonstrate their comparative advantages – in what technology/devices that the two advantages are simultaneously needed to substantially improve the efficiency/performance of the devices. While it is ideal for the authors to have this demonstration in the paper, I am of the opinion that even without the demonstration, the paper has enough new results to be published in Nature Comm.

Authors reply.

We thank the referee for his/her positive comments. As for his/her concerns, we understand that a demonstration of the usefulness of ultra-long and smooth wires would be optimal. We already showed that state-of-the-art electrical figures of merits (such as transconductance and electron mobility) can be reached with relatively small effort and we hope this may trigger further developments in the near future. In the Conclusion section of the new version of the paper, we modified this sentence:

“Since the proposed approach is very general, it can be adapted to tune the Si wires aspect ratio by choosing suitable UT-SOI and pattern periodicity combined with more complex, connected nano-architectures towards a full exploitation of their record length and atomically smooth facets.”